# Shared and modality-specific brain regions that mediate auditory and visual word comprehension

Anne Keitel[1,2]*, Joachim Gross[2,3], Christoph Kayser[4]

[1]Psychology, University of Dundee, Dundee, United Kingdom; [2]Institute of Neuroscience and Psychology, University of Glasgow, Glasgow, United Kingdom; [3]Institute for Biomagnetism and Biosignalanalysis, University of Münster, Münster, Germany; [4]Department for Cognitive Neuroscience, Faculty of Biology, Bielefeld University, Bielefeld, Germany

**Abstract** Visual speech carried by lip movements is an integral part of communication. Yet, it remains unclear in how far visual and acoustic speech comprehension are mediated by the same brain regions. Using multivariate classification of full-brain MEG data, we first probed where the brain represents acoustically and visually conveyed word identities. We then tested where these sensory-driven representations are predictive of participants' trial-wise comprehension. The comprehension-relevant representations of auditory and visual speech converged only in anterior angular and inferior frontal regions and were spatially dissociated from those representations that best reflected the sensory-driven word identity. These results provide a neural explanation for the behavioural dissociation of acoustic and visual speech comprehension and suggest that cerebral representations encoding word identities may be more modality-specific than often upheld.

*For correspondence:
a.keitel@dundee.ac.uk

Competing interests: The authors declare that no competing interests exist.

## Introduction

Acoustic and visual speech signals are both elemental for everyday communication. While acoustic speech consists of temporal and spectral modulations of sound pressure, visual speech consists of movements of the mouth, head, and hands. Movements of the mouth, lips and tongue in particular provide both redundant and complementary information to acoustic cues (*Hall et al., 2005*; *Peelle and Sommers, 2015*; *Plass et al., 2019*; *Summerfield, 1992*), and can help to enhance speech intelligibility in noisy environments and in a second language (*Navarra and Soto-Faraco, 2007*; *Sumby and Pollack, 1954*; *Yi et al., 2013*). While a plethora of studies have investigated the cerebral mechanisms underlying speech in general, we still have a limited understanding of the networks specifically mediating visual speech perception, that is lip reading (*Bernstein and Liebenthal, 2014*; *Capek et al., 2008*; *Crosse et al., 2015*). In particular, it remains unclear whether visual speech signals are largely represented in dedicated regions, or whether these signals are encoded by the same networks that mediate auditory speech perception.

Behaviourally, our ability to understand acoustic speech seems to be independent from our ability to understand visual speech. In the typical adult population, performance in auditory/verbal and visual speech comprehension tasks are uncorrelated (*Conrad, 1977*; *Jeffers and Barley, 1980*; *Mohammed et al., 2006*; *Summerfield, 1991*; *Summerfield, 1992*). Moreover, large inter-individual differences in lip reading skills contrast with the low variability seen in auditory speech tests (*Summerfield, 1992*). In contrast to this behavioural dissociation, neuroimaging and neuroanatomical studies have suggested the convergence of acoustic and visual speech information in specific brain regions (*Calvert, 1997*; *Campbell, 2008*; *Ralph et al., 2017*; *Simanova et al., 2014*). Prevalent models postulate a fronto-temporal network mediating acoustic speech representations, comprising

a word-meaning pathway from auditory cortex to inferior frontal areas, and an articulatory pathway that extends from auditory to motor regions (*Giordano et al., 2017*; *Giraud and Poeppel, 2012*; *Gross et al., 2013*; *Hickok, 2012*; *Huth et al., 2016*; *Morillon et al., 2019*). Specifically, a number of anterior-temporal and frontal regions have been implied in implementing a-modal semantic representations (*MacSweeney et al., 2008*; *Ralph et al., 2017*; *Simanova et al., 2014*) and in enhancing speech perception in adverse environments, based on the combination of acoustic and visual signals (*Giordano et al., 2017*).

Yet, when it comes to representing visual speech signals themselves, our understanding becomes much less clear. That is, we know relatively little about which brain regions mediate lip reading. Previous studies have shown that visual speech activates ventral and dorsal visual pathways and bilateral fronto-temporal circuits (*Bernstein and Liebenthal, 2014*; *Calvert, 1997*; *Campbell, 2008*; *Capek et al., 2008*). Some studies have explicitly suggested that auditory regions are also involved in lip reading (*Calvert, 1997*; *Calvert and Campbell, 2003*; *Capek et al., 2008*; *Lee and Noppeney, 2011*; *Pekkola et al., 2005*), for example by receiving signals from visual cortices that can be exploited to establish coarse-grained acoustic representations (*Bourguignon et al., 2020*). While these findings can be seen to suggest that largely the same brain regions represent acoustic and visual speech, neuroimaging studies have left the nature and the functional specificity of these visual speech representations unclear (*Bernstein and Liebenthal, 2014*; *Crosse et al., 2015*; *Ozker et al., 2018*). This is in part because most studies focused on mapping activations rather than specific semantic or lexical speech content. Indeed, alternative accounts have been proposed, which hold that visual and auditory speech representations are largely distinct (*Bernstein and Liebenthal, 2014*; *Evans et al., 2019*).

When investigating how speech is encoded in the brain, it is important to distinguish purely stimulus driven neural activity (e.g. classic 'activation') from activity specifically representing a stimulus and contributing to the participant's percept on an individual trial (*Bouton et al., 2018*; *Grootswagers et al., 2018*; *Keitel et al., 2018*; *Panzeri et al., 2017*; *Tsunada et al., 2016*). That is, it is important to differentiate the representations of sensory inputs per se from those representations of sensory information that directly contribute to, or at least correlate with, the single-trial behavioural outcome. Recent neuroimaging studies have suggested that those cerebral representations representing the physical speech are partly distinct from those reflecting the actually perceived meaning. For example, syllable identity can be decoded from temporal, occipital and frontal areas, but only focal activity in the inferior frontal gyrus (IFG) and posterior superior temporal gyrus (pSTG) mediates perceptual categorisation (*Bouton et al., 2018*). Similarly, the encoding of the acoustic speech envelope is seen widespread in the brain, but correct word comprehension correlates only with focal activity in temporal and motor regions (*Scott, 2019*; *Keitel et al., 2018*). In general, activity in lower sensory pathways seems to correlate more with the actual physical stimulus, while activity in specific higher-tier regions correlates with the subjective percept (*Crochet et al., 2019*; *Romo et al., 2012*). However, this differentiation poses a challenge for data analysis, and studies on sensory perception are only beginning to address this systematically (*Grootswagers et al., 2018*; *Panzeri et al., 2017*; *Ritchie et al., 2015*).

We here capitalise on this functional differentiation of cerebral speech representations that simply reflect the physical stimulus, from those representations of the sensory inputs that correlate with the perceptual outcome, to identify the comprehension-relevant encoding of auditory and visual word identity in the human brain. That is, we ask where and to what degree comprehension-relevant representations of auditory and visual speech overlap. To this end, we exploited a paradigm in which participants performed a comprehension task based on individual sentences that were presented either acoustically or visually (lip reading), while brain activity was recorded using MEG (*Keitel et al., 2018*). We then extracted single-trial word representations and applied multivariate classification analysis geared to quantify (i) where brain activity correctly encodes the actual word identity regardless of behavioural outcome, and (ii) where the quality of the cerebral representation of word identity (or its experimentally obtained readout) is predictive of the participant's comprehension. Note that the term 'word identity' in the present study refers to the semantic, as well as the phonological form of a word (see *Figure 1—figure supplement 1* for an exploratory semantic and phonological analysis).

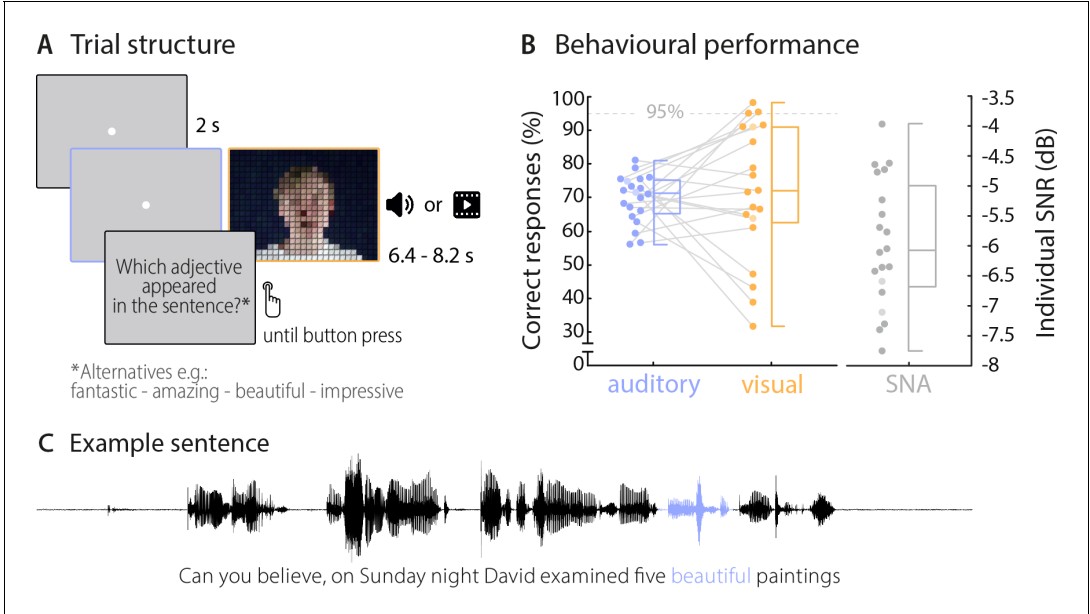

**Figure 1.** Trial structure and behavioural performance. (**A**) Trial structure was identical in the auditory and visual conditions. Participants listened to stereotypical sentences while a fixation dot was presented (auditory condition) or watched videos of a speaker saying sentences (visual condition). The face of the speaker is obscured for visualisation here only. After each trial, a prompt on the screen asked which adjective (or number) appeared in the sentence and participants chose one of four alternatives by pressing a corresponding button. Target words (here 'beautiful') occupied the 2nd or 3rd last position in the sentence. (**B**) Participants' behavioural performance in auditory (blue) and visual (orange) conditions, and their individual SNR values (grey) used for the auditory condition. Dots represent individual participants (n = 20), boxes denote median and interquartile ranges, whiskers denote minima and maxima (no outliers present). MEG data of two participants (shaded in a lighter colour) were not included in neural analyses due to excessive artefacts. Participants exceeding a performance of 95% correct (grey line) were excluded from the neuro-behavioural analysis (which was the case for three participants in the visual condition). (**C**) Example sentence with target adjective marked in blue.

The online version of this article includes the following figure supplement(s) for figure 1:

**Figure supplement 1.** Explorative representational similarity analysis (RSA) of the behavioural data (n = 20).

**Figure supplement 2.** Data preparation and classification procedures.

## Results

### Behavioural performance

On each trial, the 20 participants viewed or listened to visually or acoustically presented sentences (presented in blocks), and performed a comprehension task on a specific target word (4-alternative forced-choice identification of word identity). The 18 target words, which always occurred in the third or second last position of the sentence, each appeared in 10 different auditory and visual sentences to facilitate the use of classification-based data analysis (see table in *Supplementary file 1* for all used target words). Acoustic sentences were presented mixed with background noise, to equalise performance between visual and auditory trials. On average, participants perceived the correct target word in approximately 70% of trials across auditory and visual conditions (chance level was 25%). The behavioural performance did not differ significantly between these conditions ($M_{\text{auditory}}$ = 69.7%, SD = 7.1%, $M_{\text{visual}}$ = 71.7%, SD = 20.0%; $t(19)$ = −0.42, p=0.68; *Figure 1*), demonstrating that the addition of acoustic background noise indeed equalised performance between conditions. Still, the between-subject variability in performance was larger in the visual condition (between 31.7% and 98.3%), in line with the notion that lip reading abilities vary considerably across individuals (*Bernstein and Liebenthal, 2014*; *Summerfield, 1992*; *Tye-Murray et al., 2014*). Due to the near ceiling performance (above 95% correct), the data from three participants in the visual condition had to be excluded from the neuro-behavioural analysis. Participants also performed the task with auditory and visual stimuli presented at the same time (audiovisual condition), but because performance in this condition was near ceiling ($M_{\text{audiovisual}}$ = 96.4%, SD = 3.3%), we present the corresponding data only in the supplementary material (*Figure 2—figure supplement 2A*).

An explorative representational similarity analysis (RSA) (*Evans and Davis, 2015*; *Kriegeskorte et al., 2008*) indicated that participants' behavioural responses were influenced by both semantic and phonological features in both conditions (see Materials and methods, and *Figure 1—figure supplement 1*). A repeated-measurements ANOVA yielded a main effect of condition ($F(1,19) = 7.53$, p=0.013; mean correlations: $M_{auditory} = 0.38$, SEM = 0.01; $M_{visual} = 0.43$, SEM = 0.02) and a main effect of features ($F(1,19) = 20.98$, p<0.001, $M_{phon} = .43$, SEM = 0.01; $M_{sem} = 0.37$, SEM = 0.01). A post-hoc comparison revealed that in both conditions phonological features influenced behaviour stronger than semantic features (Wilcoxon Signed-ranks test; $Z_{auditory} = 151$, p=0.037, $Z_{visual} = 189$, p=0.002). While the small number of distinct word identities used here (n = 9 in two categories) precludes a clear link between these features and the underlying brain activity, these results suggest that participants' performance was also driven by non-semantic information.

## Decoding word identity from MEG source activity

Using multivariate classification, we quantified how well the single-trial identity of the target words (18 target words, each repeated 10 times) could be correctly predicted from source-localised brain activity ('stimulus classifier'). Classification was computed in source space at the single-subject level in a 500 ms window aligned to the onset of the respective target word. Importantly, for each trial we computed classification performance within the subset of the four presented alternative words in each trial, on which participants performed their behavioural judgement. We did this to be able to directly link neural representations of word identity with perception in a later analysis. We first quantified how well brain activity encoded the word identity regardless of behaviour ('stimulus-classification'; c.f. Materials and methods). The group-level analysis (n = 18 participants with usable MEG, cluster-based permutation statistics, corrected at p=0.001 FWE) revealed significant stimulus classification performance in both conditions within a widespread network of temporal, occipital and frontal regions (*Figure 2*).

Auditory speech was represented bilaterally in fronto-temporal areas, extending into intra-parietal regions within the left hemisphere (*Figure 2A*; *Table 1*). Cluster-based permutation statistics yielded two large clusters: a left-lateralised cluster peaking in inferior postcentral gyrus (left POST; $T_{sum} = 230.42$, p<0.001), and a right-lateralised cluster peaking in the Rolandic operculum (right RO; $T_{sum} = 111.17$, p<0.001). Visual speech was represented bilaterally in occipital areas, as well as in left parietal and frontal areas (*Figure 2B*), with classification performance between 25.9% and 33.9%. There were three clusters: a large bilateral posterior cluster that peaked in the left calcarine gyrus (left OCC; $T_{sum} = 321.78$, p<0.001), a left-hemispheric cluster that peaked in the inferior frontal gyrus (left IFG; $T_{sum} = 10.98$, p<0.001), and a left-hemispheric cluster that peaked in the postcentral gyrus (left POST; $T_{sum} = 35.83$, p<0.001). The regions representing word identity in both visual and auditory conditions overlapped in the middle and inferior temporal gyrus, the postcentral and supramarginal gyri, and the left inferior frontal gyrus (*Figure 2C*; overlap in green). MNI coordinates of cluster peaks and the corresponding classification values are given in *Table 1*. Results for the audio-visual condition essentially mirror the unimodal findings and exhibit significant stimulus classification in bilateral temporal and occipital regions (*Figure 2—figure supplement 2B*).

To directly investigate whether regions differed in their classification performance between visual and auditory conditions, we performed two analyses. First, we investigated the evidence for or against the null hypothesis of no condition difference for all grid points contributing to word classification in at least one modality (i.e. the combined clusters derived from *Figure 2A,B*). The respective Bayes factors for each ROI (from a group-level t-test) are shown in *Figure 2D*. These revealed no conclusive evidence for many grid points within these clusters ($1/3 < bf_{10} < 3$). However, both auditory clusters and the occipital visual cluster contained grid points with substantial or strong ($bf_{10} > 3$ and $bf_{10} > 10$, respectively) evidence for a significant modality difference. In contrast, the visual postcentral region (POST), the IFG and the overlap region contained many grid points with substantial evidence for no difference between modalities ($1/10 < bf_{10} < 1/3$). Second, we performed a full-brain cluster-based permutation test for a modality difference. The respective results, masked by the requirement of significant word classification in at least one modality, are shown in *Figure 2—figure supplement 1A*. Auditory classification was significantly better in clusters covering left and right auditory cortices, while visual classification was significantly better in bilateral visual sensory areas. Full-brain Bayes factors confirm that, apart from sensory areas exhibiting strong evidence for a

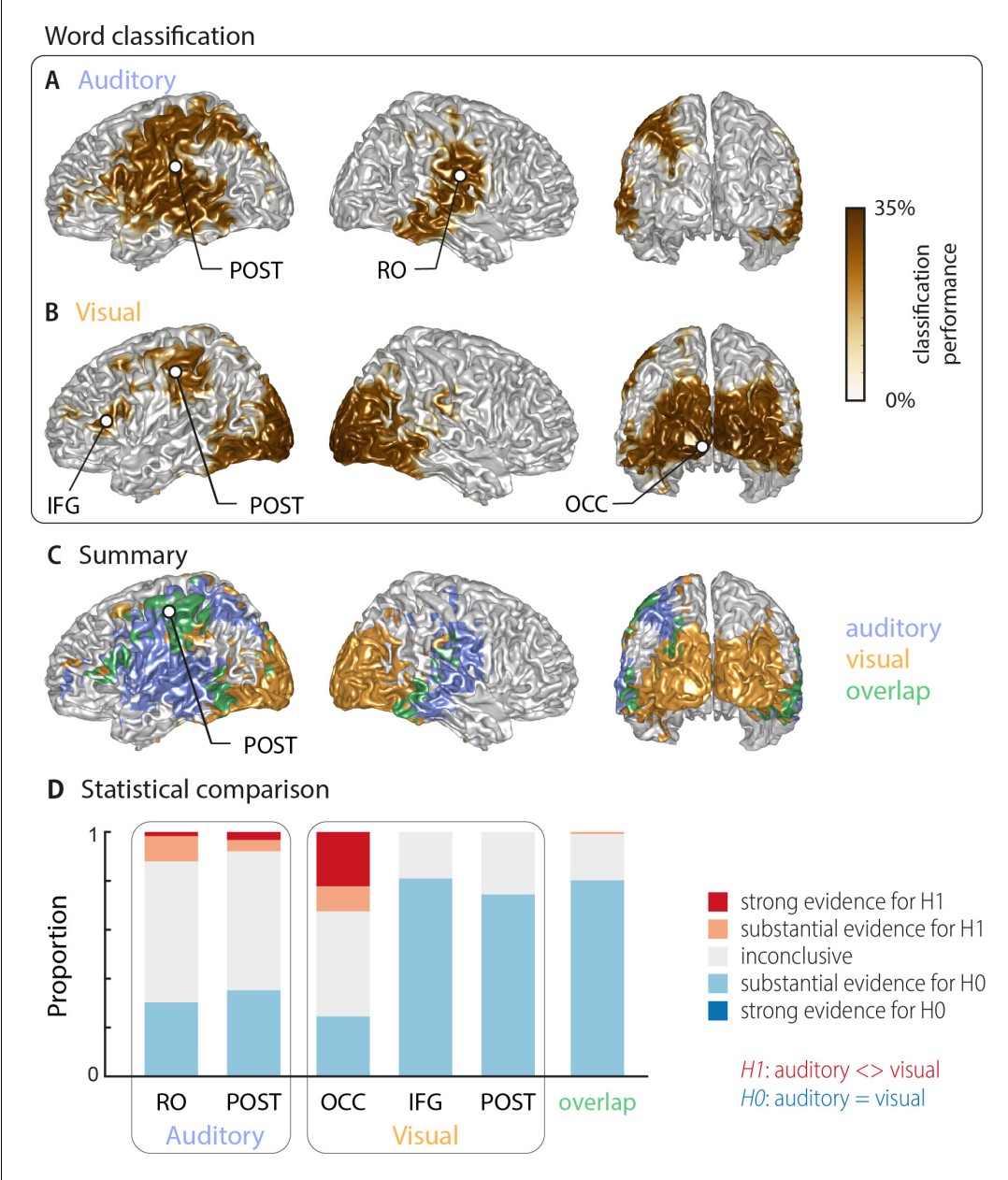

**Figure 2.** Word classification based on MEG activity regardless of behavioural performance ('stimulus classification'). Surface projections show areas with significant classification performance at the group level (n = 18; cluster-based permutation statistics, corrected at p<0.001 FWE). Results show strongest classification in temporal regions for the auditory condition (**A**) and occipital areas for the visual condition (**B**). Cluster peaks are marked with dots. Panel (**C**) overlays the significant effects from both conditions, with the overlap shown in green. The overlap contains bilateral regions in middle and inferior temporal gyrus, the inferior frontal cortex and dorsal regions of the postcentral and supramarginal gyrus (SMG). The peak of the overlap is in the postcentral gyrus. (**D**) Grid point-wise Bayes factors for a difference between auditory and visual word classification performance for all grid points in the ROIs characterised by a significant contribution to word classification in at least one modality in panel A or B (red: evidence for a difference between conditions, that is in favour of H1 [alternative hypothesis]; blue: evidence for no difference between conditions, that is in favour of H0 [null hypothesis]. RO – Rolandic Operculum; POST – postcentral gyrus; IFG – inferior frontal gyrus; OCC – occipital gyrus).

The online version of this article includes the following figure supplement(s) for figure 2:

**Figure supplement 1.** Whole-brain statistical maps for the comparison between auditory and visual word classification (*Figure 2*).

**Figure supplement 2.** Results of the audiovisual condition.

**Figure supplement 3.** Cross-classification between auditory, visual conditions and audiovisual conditions (n = 18).

**Table 1.** Peak effects of stimulus classification performance based on MEG activity.

Labels are taken from the AAL atlas (*Tzourio-Mazoyer et al., 2002*). For each peak, MNI coordinates, and classification performance (mean and SEM) are presented. Chance level for classification was 25%. Abbreviations as used in *Figure 2* are given in parentheses.

| Atlas label | MNI coordinates | | | Classification % (SEM) |
| --- | --- | --- | --- | --- |
| | X | Y | Z | |
| Auditory peaks | | | | |
| Rolandic Oper R (RO) | 41 | −14 | 20 | 28.89 (0.78) |
| Postcentral L (POST) | −48 | −21 | 25 | 29.04 (1.00) |
| Visual peaks | | | | |
| Calcarine L (OCC) | −5 | −101 | −7 | 33.92 (1.53) |
| Frontal Inf Tri L (IFG) | −48 | 23 | 1 | 26.70 (0.83) |
| Postcentral L (POST) | −51 | −24 | 47 | 26.85 (1.02) |
| Peak of overlap | | | | |
| Postcentral L (POST) | −47 | −15 | 52 | 26.50 (0.67) |

modality difference, many grid points show substantial evidence for no modality difference, or inconclusive evidence (*Figure 2—figure supplement 1B*).

Given that most clusters were found in only one hemisphere, we performed a direct test on whether these effects are indeed lateralised in a statistical sense (c.f. Materials and methods). We found evidence for a statistically significant lateralisation for both auditory clusters (left cluster peaking in POST: $t(17)$ = 5.15, $p$FDR <0.001; right cluster peaking in RO: $t(17)$ = 4.26, $p$FDR <0.01). In the visual condition, the lateralisation test for the two left clusters reached only marginal significance (left cluster peaking in IFG: $t(17)$ = 2.19, $p$FDR = 0.058; left cluster peaking in POST: $t(17)$ = 1.87, $p$FDR = 0.078). Note that the large occipital cluster in the visual condition is bilateral and we therefore did not test this for a lateralisation effect. Collectively, these analyses provide evidence that distinct frontal, occipital and temporal regions represent word identity specifically for visual and acoustic speech, while also providing evidence that regions within inferior temporal and frontal cortex, the SMG and dorsal post-central cortex reflect word identities in both modalities.

## Cerebral speech representations that are predictive of comprehension

The above analysis leaves it unclear which of these cerebral representations of word identity are actually relevant for single-trial word comprehension. That is, it remains unclear, which cerebral activations reflect the word identity in a manner that directly contributes to, or at least correlates with, participants' performance on the task. To directly address this, we computed an index of how strongly the evidence for a specific word identity in the neural single-trial word representations is predictive of the participant's response. We regressed the evidence in the cerebral classifier for word identity against the participants' behaviour (see Materials and methods). The resulting neuro-behavioural weights (regression *betas*) were converted into $t$-values for group-level analysis. The results in *Figure 3* (two-sided cluster-based permutation statistics, corrected at p=0.05 FWE) reveal several distinct regions in which neural representations of word identity are predictive of behaviour. In the auditory condition, we found five distinct clusters. Three were in the left hemisphere, peaking in the left inferior temporal gyrus (left ITG; $T_{sum}$ = 469.55, p<0.001), the inferior frontal gyrus (left IFG; $T_{sum}$ = 138.70, p<0.001), and the middle occipital gyrus (left MOG; $T_{sum}$ = 58.44, p<0.001). In the right hemisphere, the two significant clusters were in the supplementary motor area (right SMA; $T_{sum}$ = 312.48, p<0.001) and in the angular gyrus (right AG; $T_{sum}$ = 68.59, p<0.001; *Figure 3A*). In the visual condition, we found four clusters: A left-hemispheric cluster in the inferior frontal gyrus (left IFG; $T_{sum}$ = 144.15, p<0.001) and three clusters with right-hemispheric peaks, in the superior temporal gyrus (right STG; $T_{sum}$ = 168.68, p<0.001), the superior frontal gyrus (right SFG; $T_{sum}$ = 158.39, p<0.001) and the angular gyrus (right AG; $T_{sum}$ = 37.42, p<0.001; *Figure 3B*). MNI coordinates of cluster peaks and the corresponding *beta* and $t$-values are given in *Table 2*. Interestingly, these perception-relevant (i.e. predictive) auditory and visual representations did not overlap (*Figure 3C*), although some of them occurred in adjacent regions in the IFG and AG.

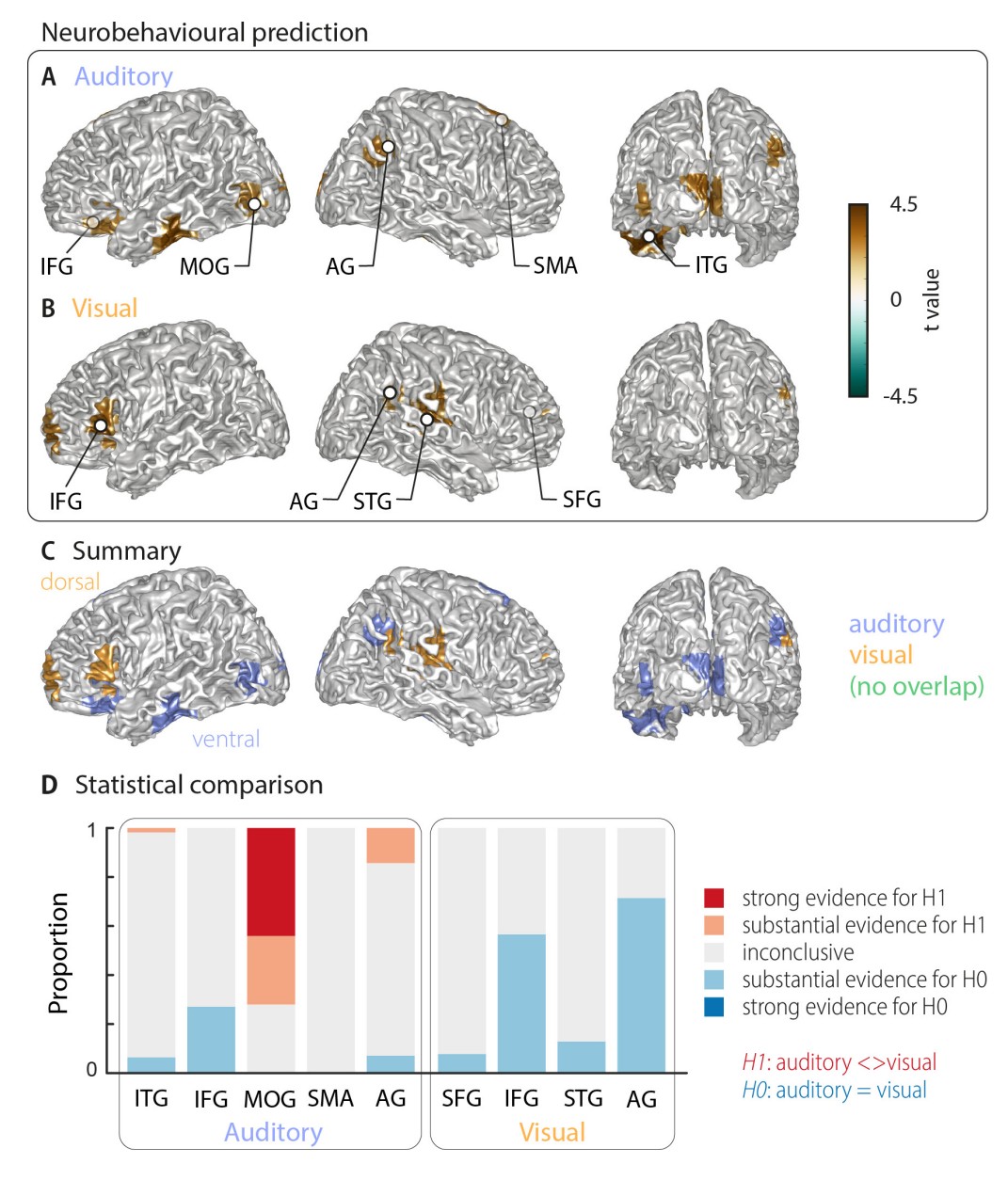

**Figure 3.** Cortical areas in which neural word representations predict participants' response. Coloured areas denote significant group-level effects (surface projection of the cluster-based permutation statistics, corrected at p<0.05 FWE). (**A**) In the auditory condition (n = 18), we found five clusters (cluster peaks are marked with dots). Three were in left ventral regions, in the inferior frontal gyrus, the inferior temporal gyrus, and the occipital gyrus, the other two were in the right hemisphere, in the angular gyrus and the supplementary motor area. (**B**) In the visual condition (n = 15; three participants were excluded due to near ceiling performance), we found four clusters: In the left (dorsal) inferior frontal gyrus, the right anterior cingulum stretching to left dorsal frontal regions, in the right angular gyrus and the right superior temporal gyrus (all peaks are marked with dots). Panel (**C**) overlays the significant effects from both conditions. There was no overlap. However, both auditory and visual effects were found in adjacent regions within the left IFG and the right AG. Panel (**D**) shows distributions of grid point-wise Bayes factors for a difference between auditory and visual conditions for these clusters (red: evidence for differences between conditions, that is in favour of H1 [alternative hypothesis]; blue: evidence for no difference between conditions, that is in favour of H0 [null hypothesis]).

The online version of this article includes the following figure supplement(s) for figure 3:

**Figure supplement 1.** Whole-brain statistical maps for the comparison between auditory and visual neurobehavioural prediction (*Figure 3*).

**Figure supplement 2.** Correlations between word classification and behavioural indices.

**Table 2.** Peak effects for the neuro-behavioural analysis.

Labels are taken from the AAL atlas (*Tzourio-Mazoyer et al., 2002*). For each local peak, MNI coordinates, regression *beta* (mean and SEM across participants) and corresponding *t*-value are presented. Abbreviations as used in *Figure 3* are given in parentheses.

| Atlas label | MNI coordinates | | | Beta (SEM) | t-value |
|---|---|---|---|---|---|
| | **X** | **Y** | **Z** | | |
| Auditory | | | | | |
| Temporal Inf L (ITG) | −41 | −23 | −26 | 0.106 (0.024) | 4.40 |
| Frontal Inf Orb L (IFG) | −28 | 25 | −9 | 0.082 (0.031) | 2.66 |
| Occipital Mid L, Occipital Inf L (MOG) | −46 | −83 | −4 | 0.079 (0.029) | 2.75 |
| Supp Motor Area R (SMA) | 3 | 11 | 52 | 0.089 (0.027) | 3.33 |
| Angular R (AG) | 49 | −67 | 40 | 0.079 (0.027) | 2.87 |
| Visual | | | | | |
| Frontal Inf Tri L (IFG) | −57 | 30 | 4 | 0.075 (0.017) | 4.34 |
| Frontal Sup Medial R, Cingulum Ant R (SFG) | 9 | 47 | 15 | 0.080 (0.028) | 2.86 |
| Temporal Sup R (STG) | 38 | −30 | 10 | 0.086 (0.023) | 3.77 |
| Angular R (AG) | 60 | −55 | 34 | 0.073 (0.020) | 3.55 |

IFG – inferior frontal gyrus; MOG – middle occipital gyrus; AG – angular gyrus; SMA – supplementary motor area; ITG – inferior temporal gyrus; IFG – inferior frontal gyrus; STG – superior temporal gyrus; SFG – superior frontal gyrus.

Again, we asked whether the behavioural relevance of these regions exhibit a significant bias towards either modality by investigating the between-condition contrast for all clusters that are significantly predictive of behaviour (Bayes factors derived from the group-level *t*-test; *Figure 3D*). Three auditory clusters contained grid points that differed substantially or strongly (bf$_{10}$ > 3 and bf$_{10}$ > 10, respectively) between modalities (left ITG, left MOG, and right AG). In addition, in two regions the majority of grid points provided substantial evidence for no difference between modalities (IFG and AG from the visual condition). A separate full-brain cluster-based permutation test (*Figure 3—figure supplement 1A*) provided evidence for a significant modality specialisation for auditory words in four clusters in the left middle occipital gyrus, left calcarine gyrus, right posterior angular gyrus, and bilateral supplementary motor area. The corresponding full-brain Bayes factors (*Figure 3—figure supplement 1B*) support this picture but also provide no evidence for a modality preference, or inconclusive results, in many other regions. Importantly, those grid points containing evidence for a significant modality difference in the full brain analysis (*Figure 3—figure supplement 1B*) correspond to those auditory ROIs derived in *Figure 3A* (MOG, posterior AG and SMA). On the other hand, regions predictive of visual word comprehension did not show a significant modality preference. Regarding the lateralisation of these clusters, we found that corresponding *betas* in the contralateral hemisphere were systematically smaller in all clusters but did not differ significantly (all $p_{FDR} \geq 0.15$), hence providing no evidence for a strict lateralisation.

To further investigate whether perception-relevant auditory and visual representations are largely distinct, we performed a cross-decoding analysis, in which we directly quantified whether the activity patterns of local speech representations are the same across modalities. At the whole-brain level, we found no evidence for significant cross-classification (two-sided cluster-based permutation statistics, neither at a corrected p=0.001 nor a more lenient p=0.05; *Figure 2—figure supplement 3A*, left panel). That significant cross-classification is possible in principle from the data, is shown by the significant results for the audiovisual condition (*Figure 2—figure supplement 3A*, right panel). An analysis of the Bayes factors for this cross-classification test confirmed that most grid points contained substantial evidence for no cross-classification between the auditory and visual conditions (*Figure 2—figure supplement 3B*, left panel). On the other hand, there was strong evidence for significant cross-classification between the uni- and multisensory conditions in temporal and occipital regions (*Figure 2—figure supplement 3B*, right panel).

## Strong sensory representations do not necessarily predict behaviour

The above results suggest that the brain regions in which sensory representations shape speech comprehension are mostly distinct from those allowing the best prediction of the actual stimulus (see *Figure 4* for direct visualisation of the results from both analyses). Only parts of the left inferior temporal gyrus (auditory condition), the right superior temporal gyrus (visual condition) and the left inferior frontal gyrus (both conditions) feature high stimulus classification and a driving role for comprehension. In other words, the accuracy by which local activity reflects the physical stimulus is generally not predictive of the impact of this local word representation on behaviour. To test this formally, we performed within-participant robust regression analyses between the overall stimulus classification performance and the perceptual weight of each local representation across all grid points. Group-level statistics of the participant-specific *beta* values provided no support for a consistent relationship between these (auditory condition: $b = 0.05 \pm 0.11$ [M ± SEM], $t(17) = 0.50$, $p_{FDR} = 0.81$; visual condition: $b = 0.03 \pm 0.11$ [M ± SEM], $t(14) = 0.25$, $p_{FDR} = 0.81$; participant-specific regression slopes are depicted in *Figure 4B*). A Bayes factor analysis also provided substantial evidence for no consistent relationship ($bf_{10} = 0.27$ and $bf_{10} = 0.27$, for auditory and visual conditions, respectively).

Still, this leaves it unclear whether variations in the strength of neural speech representations (i.e. the 'stimulus classification') can explain variations in the behavioural differences *between* participants. We therefore correlated the stimulus classification performance for all grid points with participants' behavioural data, such as auditory and lip-reading performance, and the individual SNR value. We found no significant clusters (all $ps > 0.11$, two-sided cluster-based permutation statistics, uncorrected across the four tests), indicating that stimulus classification performance was not significantly correlated with behavioural performance across participants (*Figure 3—figure supplement 2A*). The corresponding Bayes factors confirm that in the large majority of brain regions, there is substantial evidence for no correlation (*Figure 3—figure supplement 2B*).

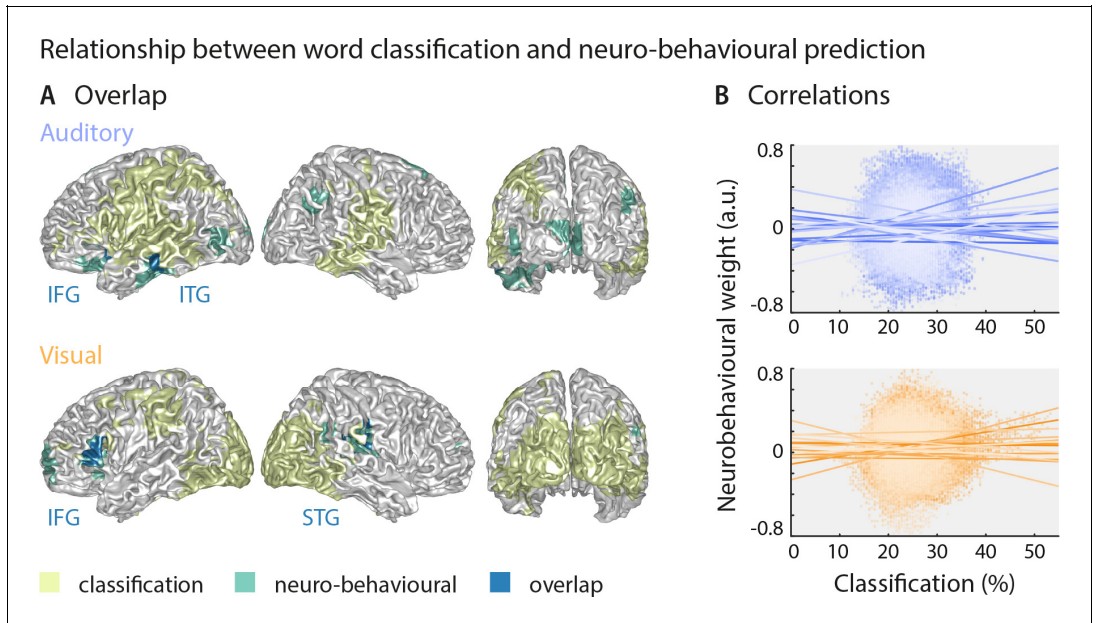

**Figure 4.** Largely distinct regions provide strong stimulus classification and mediate behavioural relevance. (**A**) Areas with significant stimulus classification (from *Figure 2*) are shown in yellow, those with significant neuro-behavioural results (from *Figure 3*) in green, and the overlap in blue. The overlap in the auditory condition (N = 14 grid points) comprised parts of the left inferior and middle temporal gyrus (ITG), and the orbital part of the left inferior frontal gyrus (IFG). The overlap in the visual condition (N = 27 grid points) comprised the triangular part of the inferior frontal gyrus (IFG), and parts of the superior temporal gyrus (STG), extending dorsally to the Rolandic operculum. (**B**) Results of a regression between word classification performance and neurobehavioural weights. Individual participant's slopes and all individual data points are shown. A group-level *t*-test on betas yielded no consistent relationship (both *t*s <0.50, both *p*s = 81). Corresponding Bayes factors (both $bf_{10}$s < 1/3) provide substantial evidence for no consistent relationship.

## Discussion

### Acoustic and visual speech are represented in distinct brain regions

Our results show that the cerebral representations of auditory and visual speech are mediated by both modality-specific and overlapping (potentially amodal) representations. While several parietal, temporal and frontal regions were engaged in the encoding of both acoustically and visually conveyed word identities ('stimulus classification'), comprehension in both sensory modalities was driven mostly by distinct networks. Only the inferior frontal and anterior angular gyrus contained adjacent regions that contributed to both auditory and visual comprehension.

This multi-level organisation of auditory and visual speech is supported by several of our findings. First, we found a partial intersection of the sensory information, where significant word classification performance overlapped in bilateral postcentral regions, inferior temporal and frontal regions and the SMG. On the other hand, auditory and visual cortices represent strongly modality-specific word identities. Second, it is also supported by the observation that anterior angular and inferior frontal regions facilitate both auditory and visual comprehension, while distinct regions support modality-specific comprehension. In particular, our data suggest that middle occipital and posterior angular representations specifically drive auditory comprehension. In addition, superior frontal and temporal regions are engaged in lip reading, although we did not find strong evidence for a modality preference of these regions. None of these comprehension-relevant regions was strictly lateralised, in line with the notion that speech comprehension is largely a bilateral process (*Kennedy-Higgins et al., 2020*).

The inability to cross-classify auditory and visual speech from local activity further supports the conclusion that the nature of local representations of acoustic and visual speech is relatively distinct. It is important to note that cross-classification probes not only the spatial overlap of two representations but also asks whether the local spatio-temporal activity patterns encoding word identity in the two sensory modalities are the same. It could be that local activity encodes a given word based on acoustic or visual evidence but using distinct activity patterns. Representations could therefore spatially overlap without using the same 'neural code'. Our results hence provide evidence that the activity patterns by which auditory and visual speech are encoded may be partly distinct, even within regions that represent both acoustically and visually mediated word information, such as the inferior frontal and anterior angular gyrus.

While we found strongest word classification performance in sensory areas, significant classification also extended into central, frontal and parietal regions. This suggests that the stimulus-domain classifier used here may also capture processes potentially related to attention or motor preparation. While we cannot rule out that levels of attention differed between conditions, we ensured by experimental design that comprehension performance did not differ between modalities. In addition, the relevant target words were placed not at the end of the sentence to prevent motor planning and preparation during their presentation (see Stimuli).

### The encoding of visual speech

The segregation of comprehension-relevant auditory and visual representations provides a possible explanation for the finding that auditory or verbal skills and visual lip reading are uncorrelated in normal-hearing adults (*Jeffers and Barley, 1980*; *Mohammed et al., 2006*; *Summerfield, 1992*). Indeed, it has been suggested that individual differences in lip reading represent something other than normal variation in speech perceptual abilities (*Summerfield, 1992*). For example, lip reading skills are unrelated to reading abilities in the typical adult population (*Arnold and Köpsel, 1996*; *Mohammed et al., 2006*), although a relationship is sometimes found in deaf or dyslexic children (*Arnold and Köpsel, 1996*; *de Gelder and Vroomen, 1998*; *Kyle et al., 2016*).

Previous imaging studies suggested that silent lip reading engages similar auditory regions as engaged by acoustic speech (*Bourguignon et al., 2020*; *Calvert, 1997*; *Calvert and Campbell, 2003*; *Capek et al., 2008*; *MacSweeney et al., 2000*; *Paulesu et al., 2003*; *Pekkola et al., 2005*), implying a direct route for visual speech into the auditory pathways and an overlap of acoustic and visual speech representations in these regions (*Bernstein and Liebenthal, 2014*). Studies comparing semantic representations from different modalities also supported large modality-independent networks (*Fairhall and Caramazza, 2013*; *Shinkareva et al., 2011*; *Simanova et al., 2014*). Yet, most

studies have focused on mapping activation strength rather than the encoding of *word identity* by cerebral speech representations. Hence, it could be that visual speech may activate many regions in an unspecific manner, without engaging specific semantic or lexical representations, maybe as a result of attentional engagement or feed-back (*Balk et al., 2013*; *Ozker et al., 2018*). Support for this interpretation comes from lip reading studies showing that auditory cortical areas are equally activated by visual words and pseudo-words (*Calvert, 1997*; *Paulesu et al., 2003*), and studies demonstrating cross-modal activations in early sensory regions also for simplistic stimuli (*Ferraro et al., 2020*; *Ibrahim et al., 2016*; *Petro et al., 2017*).

Our results suggest that visual speech comprehension is mediated by parietal and inferior frontal regions that likely contribute to both auditory and visual speech comprehension, but also engage superior temporal and superior frontal regions. Thereby our results support a route of visual speech into auditory cortical and temporal regions but provide no evidence for an overlap of speech representations in the temporal lobe that would facilitate both lip-reading and acoustic speech comprehension, in contrast to recent suggestions from a lesion-based approach (*Hickok et al., 2018*).

Two specific regions mediating lip-reading comprehension were the IFG and the anterior angular gyrus. Our results suggest that these facilitate both auditory and visual speech comprehension, in line with previous suggestions (*Simanova et al., 2014*). Previous work has also implicated these regions in the visual facilitation of auditory speech-in-noise perception (*Giordano et al., 2017*) and lip-reading itself (*Bourguignon et al., 2020*). Behavioural studies have shown that lip-reading drives the improvement of speech perception in noise (*Macleod and Summerfield, 1987*), hence suggesting that the representations of visual speech in these regions may be central for hearing in noisy environments. Interestingly, these regions resemble the left-lateralised dorsal pathway activated in deaf signers when seeing signed verbs (*Emmorey et al., 2011*). Our results cannot directly address whether these auditory and visual speech representations are the same as those that mediate the multisensory facilitation of speech comprehension in adverse environments (*Bishop and Miller, 2009*; *Giordano et al., 2017*). Future work needs to directly contrast the degree of which multisensory speech representations overlap locally to the ability of these regions to directly fuse this information.

## Cross-modal activations in visual cortex

We also found that acoustic comprehension was related to occipital brain activity (c.f. *Figure 3*). Previous work has shown that salient sounds activate visual cortices (*Feng et al., 2014*; *McDonald et al., 2013*), with top-down projections providing visual regions with semantic information, for example about object categories (*Petro et al., 2017*; *Revina et al., 2018*). The acoustic speech in the present study was presented in noise, and performing the task hence required attentional effort. Attention may therefore have automatically facilitated the entrance of top-down semantic information into occipital regions that, in a multisensory context, would encode the lip-movement trajectory, in order to maximise task performance (*McDonald et al., 2013*). The lack of significant cross-classification performance suggests that the nature of this top-down induced representation differs from that induced by direct lip-movement information.

## Sub-optimal sensory representations contribute critically to behaviour

To understand which cerebral representations of sensory information guide behaviour, it is important to dissociate those that mainly correlate with the stimulus from those that encode sensory information and guide behavioural choice. At the single neuron level some studies have proposed that only those neurons encoding the specific stimulus optimally are driving behaviour (*Britten et al., 1996*; *Pitkow et al., 2015*; *Purushothaman and Bradley, 2005*; *Tsunada et al., 2016*), while others suggest that 'plain' sensory information and sensory information predictive of choice can be decoupled across neurons (*Runyan et al., 2017*). Theoretically, these different types of neural representations can be dissected by considering the intersection of brain activity predictive of stimulus and choice (*Panzeri et al., 2017*), that is, the neural representations that are informative about the sensory environment and are used to guide behaviour. While theoretically attractive, this intersection is difficult to quantify for high-dimensional data, in part as direct estimates of this intersection, for example based on information-theoretic approaches, are computationally costly (*Pica et al., 2017*). Hence, in the past most studies, also on speech, have focused on either studying sensory encoding (e.g. by

classifying stimuli), or behaviourally predictive activity only (e.g. by classifying responses). However, the former type of cerebral representation may not guide behaviour at all, while the latter may also capture brain activity that drives perceptual errors due to intrinsic fluctuations in sensory pathways, the decision process, or even noise in the motor system (*Grootswagers et al., 2018*).

To directly quantify where auditory or visual speech is represented and this representation is used to guide comprehension we capitalised on the use of a stimulus-classifier to first pinpoint brain activity carrying relevant word-level information and to then test where the quality of the single-trial word representation is predictive of participants' comprehension (*Cichy et al., 2017*; *Grootswagers et al., 2018*; *Ritchie et al., 2015*). This approach directly follows the idea to capture processes related to the encoding of external (stimulus-driven) information and to then ask whether these representations correlate over trials with the behavioural outcome or report. Although one has to be careful in interpreting this as causally driving behaviour, our results reveal that brain regions allowing for a sub-optimal read-out of the actual stimulus are predictive of the perceptual outcome, whereas those areas allowing the best read-out not necessarily predict behaviour. This dissociation is emerging in several recent studies on the neural basis underlying perception (*Bouton et al., 2018*; *Grootswagers et al., 2018*; *Hasson et al., 2007*; *Keitel et al., 2018*). Importantly, it suggests that networks mediating speech comprehension can neither be understood by mapping speech representations during passive perception nor during task performance, if the analysis itself is not geared towards directly revealing the perception-relevant representations.

On a technical level, it is important to keep in mind that the insights derived from any classification analysis are limited by the quality of the overall classification performance. Classification performance was highly significant and reached about 10% above the respective chance level, a number that is in accordance with other neuroimaging studies on auditory pathways (*Bednar et al., 2017*; *Correia et al., 2015*). Yet, more refined classification techniques, or data obtained using significantly larger stimulus sets and more repetitions of individual target words may be able to provide even more refined insights. In addition, by design of our experiment (four response options) and data analysis, the neurobehavioral analysis was primary driven by trials in which the respective brain activity encoded the sensory stimulus correctly. We cannot specifically link the incorrect encoding of a stimulus with behaviour. This is in contrast to studies using only two stimulus or response options, where evidence for one option directly provides evidence against the other (*Frühholz et al., 2016*; *Petro et al., 2013*).

One factor that may shape the behavioural relevance of local sensory representations is the specific task imposed (*Hickok and Poeppel, 2007*). In studies showing the perceptual relevance of optimally encoding neurons, the tasks were mostly dependent on low-level features (*Pitkow et al., 2015*; *Tsunada et al., 2016*), while studies pointing to a behavioural relevance of high level regions were relying on high-level information such as semantics or visual object categories (*Grootswagers et al., 2018*; *Keitel et al., 2018*). One prediction from our results is therefore that if the nature of the task was changed from speech comprehension to an acoustic task, the perceptual relevance of word representations would shift from left anterior regions to strongly word encoding regions in the temporal and supramarginal regions. Similarly, if the task would concern detecting basic kinematic features of the visual lip trajectory, activity within early visual cortices tracking the stimulus dynamics should be more predictive of behavioural performance (*Di Russo et al., 2007*; *Keitel et al., 2019*; *Tabarelli et al., 2020*). This suggests that a discussion of the relevant networks underlying speech perception should always be task-focused.

## Conclusion

These results suggest that cerebral representations of acoustic and visual speech might be more modality-specific than often assumed and provide a neural explanation for why acoustic speech comprehension is a poor predictor of lip-reading skills. Our results also suggest that those cerebral speech representations that directly drive comprehension are largely distinct from those best representing the physical stimulus, strengthening the notion that neuroimaging studies need to more specifically quantify the cerebral mechanisms driving single-trial behaviour.

## Materials and methods

Part of the dataset analysed in the present study has been used in a previous publication (*Keitel et al., 2018*). The data analysis performed here is entirely different from the previous work and includes unpublished data.

### Participants and data acquisition

Twenty healthy, native volunteers participated in this study (nine female, age 23.6 ± 5.8 y [*M* ± *SD*]). The sample size was set based on previous recommendations (*Bieniek et al., 2016*; *Poldrack et al., 2017*; *Simmons et al., 2011*). MEG data of two participants had to be excluded due to excessive artefacts. Analysis of MEG data therefore included 18 participants (seven female), whereas the analysis of behavioural data included 20 participants. An exception to this is the neurobehavioral analysis in the visual condition, where three participants performed at ceiling and had to be excluded (resulting in n = 15 participants in *Figure 3B*). All participants were right-handed (Edinburgh Handedness Inventory; *Oldfield, 1971*), had normal hearing (Quick Hearing Check; *Koike et al., 1994*), and normal or corrected-to-normal vision. Participants had no self-reported history of neurological or language disorders. All participants provided written informed consent prior to testing and received monetary compensation of £10/h. The experiment was approved by the ethics committee of the College of Science and Engineering, University of Glasgow (approval number 300140078), and conducted in compliance with the Declaration of Helsinki.

MEG was recorded with a 248-magnetometers, whole-head MEG system (MAGNES 3600 WH, 4-D Neuroimaging) at a sampling rate of 1 KHz. Head positions were measured at the beginning and end of each run, using five coils placed on the participants' head. Coil positions were co-digitised with the head-shape (FASTRAK, Polhemus Inc, VT, USA). Participants sat upright and fixated a fixation point projected centrally on a screen. Visual stimuli were displayed with a DLP projector at 25 frames/second, a resolution of 1280 × 720 pixels, and covered a visual field of 25 × 19 degrees. Sounds were transmitted binaurally through plastic earpieces and 370 cm long plastic tubes connected to a sound pressure transducer and were presented stereophonically at a sampling rate of 22,050 Hz. Stimulus presentation was controlled with Psychophysics toolbox (*Brainard, 1997*) for MATLAB (The MathWorks, Inc) on a Linux PC.

### Stimuli

The experiment featured three conditions: auditory only (A), visual only (V), and a third condition in which the same stimulus material was presented audiovisually (AV). This condition could not be used for the main analyses as participants performed near ceiling level in the behavioural task (correct trials: *M* = 96.5%, SD = 3.4%; see *Figure 2—figure supplement 2A* for results). The stimulus material consisted of 180 sentences, based on a set of 18 target words derived from two categories (nine numbers and nine adjectives), each repeated 10 times in a different sentence. Sentences were spoken by a trained, male, native British actor. Sentences were recorded with a high-performance camcorder (Sony PMW-EX1) and external microphone. The speaker was instructed to speak clearly and naturally. Each sentence had the same linguistic structure (*Keitel et al., 2018*). An example is: '*Did you notice* (filler phase), *on Sunday night* (time phrase) *Graham* (name) *offered* (verb) *ten* (number) *fantastic* (adjective) *books* (noun)". In total, 18 possible names, verbs, numbers, adjectives, and nouns were each repeated ten times. Sentence elements were re-combined within a set of 180 sentences. As a result, sentences made sense, but no element could be semantically predicted from the previous material. To measure comprehension performance, a target word was selected that was either the adjective in one set of sentences ('fantastic' in the above example) or the number in the other set (for example, 'thirty-two'). These were always the second or third last word in a sentence, ensuring that the cerebral processes encoding these words were independent from the behavioural (motor) response. All adjective target words had a positive valence (*Scott et al., 2019*; see table in *Supplementary file 1* for all possible target words). The duration of sentences ranged from 4.2 s to 6.5 s (5.4 ± 0.4 s [*M* ± *SD*]). Noise/video onset and offset was approximately 1 s before and after the speech, resulting in stimulus lengths of 6.4 s to 8.2 s (*Figure 1*). The durations of target words ranged from 419 ms to 1,038 ms (679 ± 120 ms [*M* ± *SD*]). After the offset of the target words, the stimulus continued for 1.48 s to 2.81 s (1.98 ± 0.31 s [*M* ± *SD*]) before the end of the sentence.

The acoustic speech was embedded in noise to match performance between auditory and visual conditions. The noise consisted of ecologically valid, environmental sounds (traffic, car horns, talking), combined into a uniform mixture of 50 different background noises. The individual noise level for each participant was determined with a one-up-three-down (3D1U) staircase procedure that targets the 79.4% probability correct level (*Karmali et al., 2016*). For the staircase procedure, only the 18 possible target words (i.e. adjectives and numbers) were used instead of whole sentences. Participants were presented with a single target word embedded in noise and had to choose between two alternatives. Note that due to the necessary differences between staircase procedure (single words and two-alternative-forced-choice) and behavioural experiment (sentences and four-alternative forced-choice), the performance in the behavioural task was lower than 79.4%. The signal-to-noise ratio across participants ranged from −7.75 dB to −3.97 dB (−5.96 ± 1.06 dB [$M \pm SD$]; see *Figure 1B*).

## Experimental design

The 180 sentences were each presented in three conditions (A, V, AV), each consisting of four blocks with 45 sentences each. In each block, participants either reported the comprehended adjective or number, resulting in two 'adjective blocks' and two 'number blocks'. The order of sentences and blocks was randomised for each participant. The first trial of each block was a 'dummy' trial that was discarded for subsequent analysis; this trial was repeated at the end of the block.

During the presentation of the sentence, participants fixated either a dot (auditory condition) or a small cross on the speaker's mouth (see *Figure 1* for depiction of trial structure). After each sentence, participants were presented with four target words (either adjectives or written numbers) on the screen and had to indicate which one they perceived by pressing one of four buttons on a button box. After 2 s, the next trial started automatically. Each block lasted approximately 10 min. The two separate sessions were completed within one week.

## MEG pre-processing

Pre-processing of MEG data was carried out in MATLAB (The MathWorks, Inc) using the Fieldtrip toolbox (*Oostenveld et al., 2011*). All experimental blocks were pre-processed separately. Single trials were extracted from continuous data starting 2 s before sound/video onset and until 10 s after onset. MEG data were denoised using a reference signal. Known faulty channels ($N$ = 7) were removed before further pre-processing. Trials with SQUID jumps (on average 3.86% of trials) were detected and removed using Fieldtrip procedures with a cut-off $z$-value of 30. Before further artifact rejection, data were filtered between 0.2 and 150 Hz (fourth order Butterworth filters, forward and reverse) and down-sampled to 300 Hz. Data were visually inspected to find noisy channels (4.95 ± 5.74 on average across blocks and participants) and trials (0.60 ± 1.24 on average across blocks and participants). There was no indication for a statistical difference between the number of rejected channels or trials between conditions (two-sided $t$-tests; p>0.48 for channels, p>0.40 for trials). Finally, heart and eye movement artifacts were removed by performing an independent component analysis with 30 principal components (2.5 components removed on average). Data were further down-sampled to 150 Hz and bandpass-filtered between 0.8 and 30 Hz (fourth order Butterworth filters, forward and reverse).

## Source reconstruction

Source reconstruction was performed using Fieldtrip, SPM8, and the Freesurfer toolbox. We acquired T1-weighted structural magnetic resonance images (MRIs) for each participant. These were co-registered to the MEG coordinate system using a semi-automatic procedure (*Gross et al., 2013*; *Keitel et al., 2017*). MRIs were then segmented and linearly normalised to a template brain (MNI space). A forward solution was computed using a single-shell model (*Nolte, 2003*). We projected sensor-level timeseries into source space using a frequency-specific linear constraint minimum variance (LCMV) beamformer (*Van Veen et al., 1997*) with a regularisation parameter of 7% and optimal dipole orientation (singular value decomposition method). Covariance matrices for source were based on the whole length of trials (*Brookes et al., 2008*). Grid points had a spacing of 6 mm, resulting in 12,337 points covering the whole brain. For subsequent analyses, we selected grid

points that corresponded to cortical regions only (parcellated using the AAL atlas; *Tzourio-Mazoyer et al., 2002*). This resulted in 6490 grid points in total.

Neural time series were spatially smoothed (*Gross et al., 2013*) and normalised in source space. For this, the band-pass filtered time series for the whole trial (i.e. the whole sentence) were projected into source space and smoothed using SPM8 routines with a Full-Width Half Max (FWHM) value of 3 mm. The time series for each grid point and trial was then *z*-scored.

## Classification analysis

We used multi-variate single-trial classification to localise cerebral representations of the target words in source activity (*Grootswagers et al., 2017*; *Guggenmos et al., 2018*). Each target word was presented in ten different trials per condition. We extracted the 500 ms of activity following the onset of each target word and re-binned the source activity at 20 ms resolution. This choice of the analysis time window as made based on the typical duration of target words ($M$ = 679 ms length, see **Stimuli**). Because the words following the target word differed in each sentence, choosing a longer window would have contaminated the specific classification of target word identity. We therefore settled on a 500 ms window, which has been shown to be sufficient for word decoding (*Chan et al., 2011*) and does not include the beginning of the following word in most sentences (94%). Importantly, this analysis window did not capture post-sentence or repose periods. Classification was performed on spatial searchlights of 1.2 cm radius. The typical searchlight contained 31 neighbours (median value), with 95% of searchlights containing 17 to 33 grid points. The (leave-one-trial-out) classifier computed, for a given trial, the Pearson correlation of the spatio-temporal searchlight activity in this test-trial with the activities for the same word in all other trials (within-target distances), and with the activities of the three alternative words in all trials (between-target distances). That is, each trial was classified within the sub-set of words that was available to the participant as potential behavioural choices (see *Figure 1—figure supplement 2* for illustration). We then averaged correlations within the four candidate words and decoded the target trial as the word identity with the strongest average correlation (that is, smallest classifier distance). This classification measure is comparable to previous studies probing how well speech can be discriminated based on patterns of dynamic brain activity (*Luo and Poeppel, 2007*; *Rimmele et al., 2015*). Classification performance was averaged across blocks with numbers and adjectives as task-relevant words. For cross-condition classification (*Figure 2—figure supplement 3*), we classified the single-trial activity from the auditory (visual) condition against all trials with the same word alternatives from the other condition, or from the audiovisual condition.

## Selection of parameters and classifier procedures

We initially tested a number of different classifiers, including linear-discriminant and diagonal-linear classifiers, and then selected a correlation-based nearest-neighbour classifier as this performed slightly better than the others (although we note that the difference in peak classification performance was only on the range of 2–3% between different classifiers). We focussed on linear classifiers here because these have been shown to often perform equally well than more complex non-linear classifiers, while also offering insights that are more readily interpretable (*Haxby et al., 2014*; *Kamitani and Tong, 2005*; *Ritchie et al., 2019*).

To assess the impact of the temporal binning of MEG activity, we probed classification performance based on bins of 3.3, 20, 40 and 60 ms length. Classification performance dropped slightly when sampling the data at a resolution lower than 20 ms, particularly for auditory classification (for 3.3, 20, 40 and 60 ms bins, the mean performance of the 10% grid points with the highest values in the benchmark 20 ms classification was: auditory, 27.19 ± 0.48%, 26.81 ± 0.86%, 26.54 ± 1.00% and 25.92 ± 0.85%; visual, 28.71 ± 1.55%, 28.68 ± 1.73%, 28.54 ± 1.68% and 27.85 ± 2.04% [M ± SD]).

We also probed the influence of the spatial searchlight by (i) including each neighbouring spatial grid point into the searchlight, or (ii) averaging across grid points, and (iii) by not including a searchlight altogether. Ignoring the spatial pattern by averaging grid points led to a small drop in classification performance (individual vs average grid points: auditory, 27.19 ± 0.48 vs 26.72 ± 0.67; visual, 28.71 ± 1.55 vs 27.71 ± 1.25 [M ± SD]). Performance also dropped slightly when no searchlight was included (auditory, 26.77 ± 1.93; visual, 27.86 ± 2.50 [M ± SD]).

For the main analysis, we therefore opted for a classifier based on the MEG source data represented as spatial searchlight including each grid point within a 1.2 cm radius, and binned at 20 ms resolution.

## Quantifying the behavioural relevance of speech representations

To quantify the degree to which the classifier evidence obtained from local speech representations in favour of a specific word identity is predictive of participants' comprehension, we extracted an index of how well the classifier separated the correct word identity from the three false alternatives: the distance of the single trial classifier evidence to a decision bound (*Cichy et al., 2017*; *Grootswagers et al., 2018*; *Ritchie et al., 2015*). This representational distance was defined as the average correlation with trials of the same (within-target distances) word identity minus the mean of the correlation with the three alternatives (between-target distances; see *Figure 1—figure supplement 2*). If a local cerebral representation allows a clear and robust classification of a specific word identity, this representational distance would be large, while if a representation allows only for poor classification, or mis-classifies a trial, this distance will be small or negative. We then quantified the statistical relation between participants performance (accuracy) and these single-trial representational distances (*Cichy et al., 2017*; *Grootswagers et al., 2018*; *Panzeri et al., 2017*; *Pica et al., 2017*; *Ritchie et al., 2015*). This analysis was based on a regularised logistic regression (*Parra et al., 2005*), which was computed across all trials per participant. To avoid biasing, the regression model was computed across randomly selected subsets of trials with equal numbers of correct and wrong responses, averaging betas across 50 randomly selected trials. The resulting *beta* values were averaged across blocks with numbers and adjectives as targets and were entered into a group-level analysis. Given the design of the task (four response options, around 70% correct performance), this analysis capitalises on the relation between correctly encoded words (positive representational distance) and their relation to performance. Conversely, it is not directly able to capture how a wrongly encoded word identity relates to performance.

## Quantifying the role of phonological and semantic features to perception

For each pair of words we computed their phonological distance using the Phonological Corpus Tools (V1.4.0) based on the phonetic string similarity ('phonological edit distance') derived from the transcription tier, using the Irvine Phonotactic Online Dictionary (*Vaden et al., 2009*). We also computed pairwise semantic distances using the fastTExt vector representation of English words trained on *Common Crawl* and *Wikipedia* obtained online (file cc.en.300.vec) (*Grave et al., 2018*). The individual word vectors (300 dimensions) were length-normalised and cosine distances were computed. For each participant, we obtained a behavioural representational dissimilarity matrix (RDM) as the pair-wise behavioural confusion matrix from their behavioural data. We then implemented a representational similarity analysis (RSA) (*Kriegeskorte et al., 2008*) between phonological (semantic) representations and participants' performance. Specifically, behavioural and semantic (phonetic) RDMs were compared using Spearman's rank correlation. The resulting correlations were *z*-scored and averaged across adjectives and numbers (see *Figure 1—figure supplement 1*).

## Statistical analyses

To test the overall stimulus classification performance, we averaged the performance per grid point across participants and compared this group-averaged value to a group-average permutation distribution obtained from 3000 within-subject permutations derived with random trial labels. Cluster-based permutation was used to correct for multiple comparisons (*Maris and Oostenveld, 2007*). Significant clusters were identified based on a first-level significance derived from the 99.95[th] percentile of the permuted distribution (family-wise error [FWE] of p=0.001), using the summed statistics ($T_{sum}$) across grid points within a cluster, and by requiring a minimal cluster size of 10 grid points. The resulting clusters were considered if they reached a *p*-value smaller than 0.05.

For the neuro-behavioural analyses, the regression *betas* obtained from the logistic regression were transformed into group-level *t*-values. These were compared with a surrogate distribution of *t*-values obtained from 3000 within-subject permutations using shuffled trial labels and using cluster-based permutations as above. The first-level significance threshold (at p<0.05) was determined per

condition based on the included sample size (*t*-value of *t* = 2.1 for the 18 participants in the auditory condition and *t* = 2.2 for 15 participants in the visual condition), and the resulting clusters were considered significant if they reached a p-value smaller than 0.05.

Resulting clusters were tested for lateralisation (*Liégeois et al., 2002*; *Park and Kayser, 2019*). For this, we extracted the participant-specific classification performance (or regression *betas*, respectively) for each cluster and for the corresponding contralateral grid points. These values were averaged within each hemisphere and the between-hemispheric difference was computed using a group-level, two-sided *t*-test. Resulting *p*-values were corrected for multiple comparisons by controlling the FDR at p$\leq$0.05 (*Benjamini and Hochberg, 1995*). We only use the term 'lateralised' if the between-hemispheric difference is statistically significant.

To determine whether individual local effects (e.g. stimulus classification or behavioural prediction) were specific to either condition, we implemented a direct contrast between conditions. For each grid point, we computed a group-level *t*-test. We then subjected these to the same full-brain cluster-based permutation approach as described above. In addition, we converted the group-level *t*-values to a JZS Bayes factor using a default scale factor of 0.707 (*Rouder et al., 2009*). We then quantified the number of grid points per region of interest that exhibited a specific level of evidence in favour of the null hypothesis of no effect versus the alternative hypothesis ($H_0$ vs $H_1$) (*Jeffreys, 1998*). Using previous conventions (*Wagenmakers et al., 2011*), the Bayes factors were interpreted as showing evidence for $H_1$ if they exceeded a value of 3, and evidence for $H_0$ if they were below $\frac{1}{3}$, with the intermediate range yielding inconclusive results. We also calculated Bayes factors from Pearson correlation coefficients (for a control analysis between classification performance and behavioural data), using the same conventions (*Wetzels and Wagenmakers, 2012*).

To investigate the relationship between stimulus classification and neurobehavioral results, we performed a robust linear regression within each participant for all grid points. The participant-specific *beta* values were then tested against zero using a two-sided *t*-test (*Keitel et al., 2017*).

## Acknowledgements

This research was supported by the UK Biotechnology and Biological Sciences Research Council (BBSRC, BB/L027534/1). CK is supported by the European Research Council (ERC-2014-CoG; grant No 646657); JG by the Wellcome Trust (Joint Senior Investigator Grant, No 098433), DFG (GR 2024/5-1) and IZKF (Gro3/001/19). The authors declare no competing financial interests. We are grateful to Lea-Maria Schmitt for guiding the semantic distance analysis.

## Additional information

### Funding

| Funder | Grant reference number | Author |
| --- | --- | --- |
| Biotechnology and Biological Sciences Research Council | BB/L027534/1 | Joachim Gross Christoph Kayser |
| H2020 European Research Council | ERC-2014-CoG (grant No 646657) | Christoph Kayser |
| Wellcome | Joint Senior Investigator Grant (No 098433) | Joachim Gross |
| Deutsche Forschungsgemeinschaft | GR 2024/5-1 | Joachim Gross |
| Interdisziplinäres Zentrum für Klinische Forschung, Universitätsklinikum Würzburg | Gro3/001/19 | Joachim Gross |

The funders had no role in study design, data collection and interpretation, or the decision to submit the work for publication.

## Author contributions
Anne Keitel, Conceptualization, Resources, Formal analysis, Validation, Investigation, Visualization, Methodology, Writing - original draft, Writing - review and editing; Joachim Gross, Supervision, Funding acquisition, Writing - original draft, Writing - review and editing; Christoph Kayser, Conceptualization, Resources, Formal analysis, Supervision, Funding acquisition, Investigation, Visualization, Methodology, Writing - original draft, Writing - review and editing

## Author ORCIDs
Anne Keitel ![ORCID] https://orcid.org/0000-0003-4498-0146
Christoph Kayser ![ORCID] http://orcid.org/0000-0001-7362-5704

## Ethics
Human subjects: All participants provided written informed consent prior to testing and received monetary compensation of £10/h. The experiment was approved by the ethics committee of the College of Science and Engineering, University of Glasgow (approval number 300140078), and conducted in compliance with the Declaration of Helsinki.

## Decision letter and Author response
Decision letter https://doi.org/10.7554/eLife.56972.sa1
Author response https://doi.org/10.7554/eLife.56972.sa2

# Additional files
## Supplementary files
• Supplementary file 1. Target words used in this study (9 adjectives and nine numbers, each presented in 10 different sentences). Note that adjectives were comparable with regard to their positive valence (*Scott et al., 2019*).

• Transparent reporting form

## Data availability
All relevant data and stimuli lists have been deposited to Dryad, under the https://doi.org/10.5061/dryad.zkh18937w.

The following dataset was generated:

| Author(s) | Year | Dataset title | Dataset URL | Database and Identifier |
|---|---|---|---|---|
| Keitel A, Gross J, Kayser C | 2020 | Data from: Shared and modality-specific brain regions that mediate auditory and visual word comprehension | https://doi.org/10.5061/dryad.zkh18937w | Dryad Digital Repository, 10.5061/dryad.zkh18937w |

The following previously published dataset was used:

| Author(s) | Year | Dataset title | Dataset URL | Database and Identifier |
|---|---|---|---|---|
| Keitel A, Gross J, Kayser C | 2018 | Data from: Perceptually relevant speech tracking in auditory and motor cortex reflects distinct linguistic features | https://datadryad.org/stash/dataset/doi:10.5061/dryad.1qq7050 | Dryad Digital Repository, 10.5061/dryad.1qq7050 |

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
