## [Decision Letter]

**Acceptance summary:**

We can understand speech not only through hearing the sound, but also through reading from somebody's lips. In this study, Keitel et al. show that the identity of words is encoded by similar neural networks, whether they are presented through auditory or through visual signals. However, the comprehension of speech in the two modalities involves largely different brain areas, suggesting that the neural mechanisms for auditory and for visual speech comprehension differ more than previously believed.

**Decision letter after peer review:**

Thank you for submitting your work entitled "Largely distinct networks mediate perceptually-relevant auditory and visual speech representations" for consideration by *eLife*. Your article has been reviewed by three peer reviewers, including Tobias Reichenbach as the Reviewing Editor and Reviewer #1, and the evaluation has been overseen by a Senior Editor. The following individuals involved in review of your submission have agreed to reveal their identity: Matthew H Davis (Reviewer #2); John Plass (Reviewer #3).

Our decision has been reached after consultation between the reviewers. The individual reviews below and the discussion have identified significant revisions and additional analysis that are required for the manuscript to be published in *eLife*. Because this additional work would likely take more than the two months that we allow for revisions, we regret to inform you we must formally decline your manuscript at this time.

However, we would consider a new submission in which the concerns raised below have been addressed. If you choose to submit a new version of the manuscript to *eLife*, we will make every effort for the new manuscript to be assessed by the same reviewers and Reviewing Editor.

Reviewer #1:

The authors investigate audiovisual speech processing using MEG. They present participants with speech in noise, as well as with videos of talking faces. In both conditions, subjects understand about 70% of the speech. The authors then investigate which brain areas decode word identity, as well as which brain areas predict the subject's actual comprehension. They find largely distinct areas for encoding of words as well as for predicting subject performance. Moreover, for both aspects of speech processing, the brain areas differ largely between the auditory and the visual stimulus presentation.

The paper is well written, and the obtained results shed new light on audio-visual speech processing. In particular, they show a large degree of modality-specific processing, as well as a dissociation between the brain areas for speech representation and for encoding comprehension. I am therefore in favour of publication. But I have two major comments that I would like the authors to address in a revised version.

1) I don't see the point of presenting the supplementary Figure 1. Since the behavioural results are at ceiling, only 8 subjects (those whose performance is away from the ceiling) are included in the MEG analysis. But this number of subjects is too small to draw conclusive results. The authors account for this to a degree by labelling the results as preliminary. But since the results cannot be considered conclusive, I think they should either be left out or modified to be conclusive.

2) The correlation between stimulus classification and the behavioural performance is only done for the visual condition ( – subsection “Strong sensory representations do not necessarily predict behaviour”). The authors state that this correlation can't be performed in the auditory condition because the comprehension scores there were 70%. But Figure 1B shows significant subject-to-subject variability in speech comprehension around 70%. While the variation is less than for lip reading, I don't see a reason why this performance cannot be related to the stimulus classification as well. Please add that.

Reviewer #2:

This paper describes an MEG study in which neural responses to unimodal auditory and visual (lip-read) sentences are analysed to determine the spatial locations at which brain responses provide information to distinguish between words, whether and how these responses are associated with correct perceptual identification of words, and the degree to which common neural representations in overlapping brain regions contribute to perception of auditory and visual speech.

Results show largely distinct networks of auditory, visual brain areas that convey information on word identity in heard and lip-read speech and that contribute to perception (inferred by increased information in neural representations for correctly identified words). There was some, limited neural overlap seen in higher-order language areas (such as inferior frontal gyrus, and temporal pole). However, attempts at cross-decoding – i.e. testing for common neural representations between words that were identified in visual and auditory speech – were non-significant. This is despite significant cross-decoding of both auditory and visual speech using audio-visual presentation.

Results are interpreted as showing separate, modality-specific neural representations of auditory and visual word identities, and taken to explain the independence (i.e. non-correlation) between individual differences in measures of auditory and visual speech perception abilities.

This paper addresses an important and interesting topic. Individual differences in auditory and visual speech perception are well established, and the lack of correlation between these abilities in the population appears to be a well-replicated, but currently under-explained, finding. Indeed, this basic observation goes against the dominant thrust of research on audio-visual speech perception which is largely concerned with the convergence of auditory and visual speech information. The authors' use of source-localised MEG classification analyses to address this issue is novel and the results presented build on a sufficient number of existing findings (e.g. the ability of auditory and visual responses to identify spoken and lip-read words, respectively), for me to find the two more surprising results: (1) limited overlap between audio and visual decoding, and (2) no significant cross-decoding of audio and visual speech to be intriguing and well worth publishing.

However, at the same time, I had some significant concerns that both of the key results that I have identified here depend on null findings in whole brain analyses of source-localised MEG responses. The authors must surely be aware that the absence of results reaching corrected significance cannot be taken to indicate that these effects are definitely absent. Absence of evidence is not evidence of absence, and this is particularly true when; (1) effects are tested using conventional null hypothesis significance testing, and (2) whole-brain correction for multiple comparisons are required which substantially reduce statistical sensitivity.

Only by directly subtracting statistical maps of auditory and visual word classification can the authors conclude that there is a reliable difference between the brain regions that contribute to visual and auditory word identification. Furthermore, the authors' presentation of the regression betas for peak regions from auditory and visual classification (in Figure 3D) are misleading. Given how these peaks are defined (from whole-brain search for regions showing unimodal classification), it's inevitable that maxima from auditory classification will be less reliable when tested on visual classification (and vice-versa).

These problems become particularly acute for concluding – as I think the authors wish to – that auditory areas don't contribute to word classification in visual speech. This requires confirming a null hypothesis. This can only be achieved with a Bayesian analysis which quantifies the likelihood of the null hypothesis in different brain regions. Only by performing this analysis can the authors be confident that there is not reliable classification in auditory areas that would be detected had they performed a study with greater statistical power. The authors might wish to make use of independent data – such as from the audiovisual speech condition presented in the supporting information – to define ROIs and/or expected effect sizes based on independent data.

The same problem with interpretation of null effects arises in the cross-modality decoding analyses. It is striking that these analyses are reliable for audiovisual speech but not for auditory speech. However, while this shows that the method the authors are using can detect reliable effects of a certain size, the analyses presented cannot be used to infer whether the most likely interpretation is that cross-decoding of unimodal auditory and visual speech is absent, or that it is present, but fails to reach a stringent level of significance. The authors wish to conclude that this effect is absent, they say: "The inability to cross-classify auditory and visual speech from local brain activity further supports the conclusion that acoustic and visual speech representations are largely distinct", but do not have a sound statistical basis on which to draw this conclusion.

I think some substantial additional analysis, and careful consideration of which conclusions are, or are not supported by direct statistical analyses are required if this work is to be published with the current framing. This is not to say that word identity decoding in auditory and visual speech is uninteresting, or that the seeming lack of correlation between these abilities is not of interest. Only that I found the fundamental claim in the paper – that there's no common phonological representation of auditory and visual speech – to be insufficiently supported by the data presented in the current manuscript.

Reviewer #3:

In this manuscript, Keital et al. use MEG pattern classification to compare the cortical regions involved in the encoding and comprehension of auditory and visual speech. The article is well-written, addresses a scientifically valuable topic, and extends the current literature by employing multivariate approaches. In their stimulus classification analysis, the authors found that visually-presented words were best classified on the basis of signals localized to occipital regions, while auditory words were best classified in perisylvian regions. In their neuro-behavioral decoding analysis, the authors found that classification strength predicted behavioral accuracy in largely distinct regions during auditory versus visual stimulation. They conclude that perceptually-relevant speech representations are largely modality specific.

While I largely agreed with the authors' rationale in employing these techniques, I had some concerns regarding the statistical/mathematical details of their approach. First and most importantly, the neuro-behavioral decoding analysis presented in Figure 3A and 3C does not directly test the hypothesis that forms their primary conclusion (i.e., that "Largely distinct networks mediate perceptually-relevant auditory and visual speech representations"). To test this hypothesis directly, it would be necessary to compare neuro-behavioral decoding accuracy between the auditory and visual conditions for each vertex and then multiple-comparison correct across vertices. That is, rather than comparing decoding accuracy against the null separately for each condition, decoder accuracy should be compared directly across conditions. The authors perform a similar analysis in Figure 3D, but this analysis may inflate Type-I error because it involves pre-selecting peaks identified in each single-condition analysis.

Second, throughout the manuscript, a wide variety of statistical techniques are used which are sometimes internally inconsistent, weakly or not explicitly justified, or interpreted in a manner that is not fully consistent with their mathematical implications. For example, for different analyses, different multiple comparison correction procedures are used (FDR vs. cluster-based FWE control), without a clear justification. It would be best to use similar corrections across analyses to ensure similar power. Also, the authors report that they: "Tested a number of different classifiers, […] then selected a correlation-based nearest-neighbour classifier as this performed slightly better than the others". Absent an a priori justification for the chosen approach, it would be helpful to know whether the reported results are robust to these design decisions. Finally, it is not clear to me that the neuro-behavioral decoding technique employed here is particularly well-suited for identifying regions that represent participants' percepts. It seems the most natural way to perform such an analysis would be to train a classifier to predict participants' trial-wise responses. By contrast, the technique employed here compares (binary) trial-wise accuracy with "representational distance", computed as the difference between "the average correlation with trials of the same (correct) word identity and the mean of the correlation with the three alternatives." Thus, the classifier would not be expected to identify regions with response patterns that predict participants' percepts, but those which exhibit a target-like pattern when the participant responds accurately. It is therefore perhaps best conceived of as an "accuracy classifier" rather than a "percept classifier." This may be problematic for the authors' interpretation because activity in areas unrelated to perceptual representations (e.g., areas involved in unimodal attention) could also be predictive of accuracy.

Finally, in some cases, the methods description was not self-sufficient, leaving the reader to consult references to fully understand. One critical question is how the spatial and temporal dimensions of the data were used in the classifier. If classification is primarily driven by the temporal dimension, it is not clear that successful classification really relies on pattern similarity in population responses, rather than inter-trial temporal covariation produced by, e.g., phase-resetting or entrainment to stimulus dynamics. In the cited methods articles, spatial classification is performed separately for different time bins, alleviating this concern. It would be important to critically consider this detail in interpreting these results.

[Editors’ note: further revisions were suggested prior to acceptance, as described below.]

Thank you for submitting your article "Largely distinct networks mediate perceptually-relevant auditory and visual speech representations" for consideration by *eLife*. Your article has been reviewed by three peer reviewers, including Tobias Reichenbach as the Reviewing Editor and Reviewer #1, and the evaluation has been overseen by Barbara Shinn-Cunningham as the Senior Editor. The following individual involved in review of your submission has agreed to reveal their identity: Matthew H Davis (Reviewer #2).

The reviewers have discussed the reviews with one another and the Reviewing Editor has drafted this decision to help you prepare a revised submission.

As the editors have judged that your manuscript is of interest, but as described below that additional experiments are required before it is published, we would like to draw your attention to changes in our revision policy that we have made in response to COVID-19 (https://elifesciences.org/articles/57162). First, because many researchers have temporarily lost access to the labs, we will give authors as much time as they need to submit revised manuscripts. We are also offering, if you choose, to post the manuscript to bioRxiv (if it is not already there) along with this decision letter and a formal designation that the manuscript is 'in revision at *eLife*'. Please let us know if you would like to pursue this option. (If your work is more suitable for medRxiv, you will need to post the preprint yourself, as the mechanisms for us to do so are still in development.)

Summary:

The revised paper presents a better-fitting analysis, and does a more nuanced job in discussing the results, than the original manuscript. However, there are still a few major criticisms that we have for the analysis, detailed below.

Essential revisions:

1) Brain-wide, multiple-comparison corrected tests comparing auditory versus visual decoding are still lacking. The authors have now provided vertex-wise Bayes factors within areas that showed significant decoding in each individual condition. Unfortunately, this is not satisfactory, because these statistics are (1) potentially circular because ROIs were pre-selected based on an analysis of individual conditions, (2) not multiple-comparison corrected, and (3) rely on an arbitrary prior that is not calibrated to the expected effect size. Still, ignoring these issues, the only area that appears to contain vertices with "strong evidence" for a difference in neuro-behavioral decoding is the MOG, which wouldn't really support the claim of "largely distinct networks" supporting audio vs. visual speech representation.

The authors may address these issues, for instance, by

i) presenting additional whole-brain results – e.g. for a direct comparison of auditory and visual classification (in Figure 2) and of perceptual prediction (in Figure 3).

ii) presenting voxel-wise maps of Bayesian evidence values (as in Figure 2—figure supplement 3) for the statistical comparisons shown in Figure 2D, and Figure 3D

iii) in the text included in Figure 2D and 3D making clear what hypotheses correspond to the null hypothesis and to the alternative hypothesis (i.e. auditory = visual, auditory <> visual).

2) As noted before by reviewer 3, the classifiers used in this study do not discriminate between temporal versus spatial dimensions of decoding accuracy. This leaves it unclear whether the reported results are driven by (dis)similarity of spatial patterns of activity (as in fMRI-based MVPA), temporal patterns of activity (e.g., oscillatory "tracking" of the speech signal), or some combination. As these three possibilities could lead to very different interpretations of the data, it seems critical to distinguish between them. For example, the authors write "the encoding of the acoustic speech envelope is seen widespread in the brain, but correct word comprehension correlates only with focal activity in temporal and motor regions," but, as it stands, their results could be partly driven by this non-specific entrainment to the acoustic envelope.

In their response, the authors show that classifier accuracy breaks down when spatial or temporal information is degraded, but it would be more informative to show how these two factors interact. For example, the methods article cited by the authors (Grootswagers, Wardle and Carlson, 2017) shows classification accuracy for successive time bins after stimulus onset (i.e., they train different classifiers for each time bin 0-100 ms, 100-200 ms, etc.). The timing of decoding accuracy in different areas could also help to distinguish between different plausible explanations of the results.

Finally, it is somewhat unclear how spatial and temporal information are combined in the current classifier. Figure 1—figure supplement 2 creates the impression that the time-series for each vertex within a spotlight were simply concatenated. However, this would conflate within-vertex (temporal) and across-vertex (spatial) variance.

3) The concern that the classifier could conceivably index factors influencing "accuracy" rather than the perceived stimulus does not appear to be addressed sufficiently. Indeed, the classifier is referred to as identifying "sensory representations" throughout the manuscript, when it could just as well identify areas involved in any other functions (e.g., attention, motor function) that would contribute to accurate behavioral performance. This limitation should be acknowledged in the manuscript. The authors could consider using the timing of decoding accuracy in different areas to disambiguate these explanations.

The authors state in their response that classifying based on the participant's reported stimulus (rather than response accuracy) could "possibly capture representations not related to speech encoding but relevant for behaviour only (e.g. pre-motor activity). These could be e.g. brain activity that leads to perceptual errors based on intrinsic fluctuations in neural activity in sensory pathways, noise in the decision process favouring one alternative response among four choices, or even noise in the motor system that leads to a wrong button press without having any relation to sensory representations at all."

But, it seems that all of these issues would also effect the accuracy-based classifier as well. Moreover, it seems that intrinsic fluctuations in sensory pathways, or possibly noise in the decision process, are part of what the authors are after. If noise in a sensory pathway can be used to predict particular inaccurate responses, isn't that strong evidence that it encodes behaviorally-relevant sensory representations? For example, intrinsic noise in V1 has been found to predict responses in a simple visual task in non-human primates, with false alarm trials exhibiting noise patterns that are similar to target responses (Seidemann and Geisler, 2018). Showing accurate trial-by-trial decoding of participants' incorrect responses could similarly provide stronger evidence that a certain area contributes to behavior.

---

## [Author Response]

[Editors’ note: the authors resubmitted a revised version of the paper for consideration. What follows is the authors’ response to the first round of review.]

We would like to thank the reviewers and editors for their very helpful and constructive comments. Based on these we have substantially improved the manuscript, both by refining the statistical approach and the interpretation of the results.

A main concern shared by several reviewers was that some of our conclusions were based on statistical null results, and these were not sufficiently supported by quantitative analysis (e.g. Bayes factors). With this in mind, we have revised the entire analysis pipeline. First, we ensured that all analyses are based on the same statistical corrections for multiple comparisons (which was another concern). Second, we implemented direct condition contrasts as requested and now provide Bayes factors to substantiate the evidence for or against the respective null hypotheses. Third, we implemented additional control analyses to understand whether and how spatial and temporal response patterns contribute to the overall classification performance. And fourth, we analysed semantic and phonological stimulus features to obtain a better understanding of the sensory features driving comprehension performance.

All analyses were computed de novo for the revision. Due to changes in the statistical procedures some of the specific details in the results have changed (e.g. the number of clusters emerging in specific analyses). However, the main conclusions put forward in the previous version still hold and thanks to the additional analyses we can now provide a more refined interpretation of these in the Discussion.

Reviewer #1:The authors investigate audiovisual speech processing using MEG. They present participants with speech in noise, as well as with videos of talking faces. In both conditions, subjects understand about 70% of the speech. The authors then investigate which brain areas decode word identity, as well as which brain areas predict the subject's actual comprehension. They find largely distinct areas for encoding of words as well as for predicting subject performance. Moreover, for both aspects of speech processing, the brain areas differ largely between the auditory and the visual stimulus presentation.The paper is well written, and the obtained results shed new light on audio-visual speech processing. In particular, they show a large degree of modality-specific processing, as well as a dissociation between the brain areas for speech representation and for encoding comprehension. I am therefore in favour of publication. But I have two major comments that I would like the authors to address in a revised version.1) I don't see the point of presenting the supplementary Figure 1. Since the behavioural results are at ceiling, only 8 subjects (those whose performance is away from the ceiling) are included in the MEG analysis. But this number of subjects is too small to draw conclusive results. The authors account for this to a degree by labelling the results as preliminary. But since the results cannot be considered conclusive, I think they should either be left out or modified to be conclusive.

We agree that the number of participants who performed below ceiling in the audiovisual condition is too small for conclusive results. Hence, we have removed the respective data from the manuscript but decided to keep the audiovisual results that included all participants in Figure 2—figure supplement 2. There we show the AV data regarding behavioural performance (*N* = 20) and classification performance (N = 18). We also kept the cross-classification analysis between A/V and the AV conditions to demonstrate that cross-classification is possible in principle (Figure 2—figure supplement 3).

2) The correlation between stimulus classification and the behavioural performance is only done for the visual condition (subsection “Strong sensory representations do not necessarily predict behaviour”). The authors state that this correlation can't be performed in the auditory condition because the comprehension scores there were 70%. But Figure 1B shows significant subject-to-subject variability in speech comprehension around 70%. While the variation is less than for lip reading, I don't see a reason why this performance cannot be related to the stimulus classification as well. Please add that.

We have added the respective correlation for the auditory condition, which is now presented in Figure 3—figure supplement 2A. We have also added a correlation between auditory/visual classification and individual auditory SNR values (please see response to reviewer 2 below). The statistical analysis for these correlations was adapted to be consistent with analyses in the rest of the manuscript. A cluster-based permutation test did not yield any significant results. We also report respective Bayes factors, which supported overall results (Figure 3—figure supplement 2B).

Reviewer #2:This paper describes an MEG study in which neural responses to unimodal auditory and visual (lip-read) sentences are analysed to determine the spatial locations at which brain responses provide information to distinguish between words, whether and how these responses are associated with correct perceptual identification of words, and the degree to which common neural representations in overlapping brain regions contribute to perception of auditory and visual speech.Results show largely distinct networks of auditory, visual brain areas that convey information on word identity in heard and lip-read speech and that contribute to perception (inferred by increased information in neural representations for correctly identified words). There was some, limited neural overlap seen in higher-order language areas (such as inferior frontal gyrus, and temporal pole). However, attempts at cross-decoding – i.e. testing for common neural representations between words that were identified in visual and auditory speech – were non-significant. This is despite significant cross-decoding of both auditory and visual speech using audio-visual presentation.Results are interpreted as showing separate, modality-specific neural representations of auditory and visual word identities, and taken to explain the independence (i.e. non-correlation) between individual differences in measures of auditory and visual speech perception abilities.This paper addresses an important and interesting topic. Individual differences in auditory and visual speech perception are well established, and the lack of correlation between these abilities in the population appears to be a well-replicated, but currently under-explained, finding. Indeed, this basic observation goes against the dominant thrust of research on audio-visual speech perception which is largely concerned with the convergence of auditory and visual speech information. The authors' use of source-localised MEG classification analyses to address this issue is novel and the results presented build on a sufficient number of existing findings (e.g. the ability of auditory and visual responses to identify spoken and lip-read words, respectively), for me to find the two more surprising results: (1) limited overlap between audio and visual decoding, and (2) no significant cross-decoding of audio and visual speech to be intriguing and well worth publishing.However, at the same time, I had some significant concerns that both of the key results that I have identified here depend on null findings in whole brain analyses of source-localised MEG responses. The authors must surely be aware that the absence of results reaching corrected significance cannot be taken to indicate that these effects are definitely absent. Absence of evidence is not evidence of absence, and this is particularly true when; (1) effects are tested using conventional null hypothesis significance testing, and (2) whole-brain correction for multiple comparisons are required which substantially reduce statistical sensitivity.Only by directly subtracting statistical maps of auditory and visual word classification can the authors conclude that there is a reliable difference between the brain regions that contribute to visual and auditory word identification. Furthermore, the authors' presentation of the regression betas for peak regions from auditory and visual classification (in Figure 3D) are misleading. Given how these peaks are defined (from whole-brain search for regions showing unimodal classification), it's inevitable that maxima from auditory classification will be less reliable when tested on visual classification (and vice-versa).

The reviewer points out a critical shortcoming in our previous submission: the lack of evidence for statistical null results. We have addressed this point using additional data analyses, whereby we now provide direct between-condition contrasts for all grid-points in the significant clusters, both for the stimulus classification (Figure 2) and the neuro-behavioural analysis (Figure 3). For each of these we computed a direct contrast between the A and V conditions and derived the associated Bayes factors to substantiate the evidence for or against the respective null hypothesis. These new results are presented in Figures 2D and 3D and support our conclusion that stimulus information for acoustic and visual speech is provided both by potentially amodal regions (e.g. post-central regions) as well as by regions that contain (significant) information only about a single modality (e.g. occipital regions). Furthermore, they support the conclusion that auditory and visual comprehension are driven in large by distinct networks, but also are facilitated by an overlap of auditory and visual representations in angular and frontal regions. We revised the Discussion and conclusions in the light of this refined analysis.

The reviewer also points out that our presentation of the effects obtained at local peak voxels was misleading. We did not intend to present these as an (indeed) circular statistical analysis, but rather as post-hoc visualisation and quantification of the underlying effects. In the revised manuscript, we now avoid this potentially misleading step, and derive significant clusters from the respective full-brain maps and simply report local peak results in Tables 1 and 2.

These problems become particularly acute for concluding – as I think the authors wish to – that auditory areas don't contribute to word classification in visual speech. This requires confirming a null hypothesis. This can only be achieved with a Bayesian analysis which quantifies the likelihood of the null hypothesis in different brain regions. Only by performing this analysis can the authors be confident that there is not reliable classification in auditory areas that would be detected had they performed a study with greater statistical power. The authors might wish to make use of independent data – such as from the audiovisual speech condition presented in the supporting information – to define ROIs and/or expected effect sizes based on independent data.

Following this comment, we now provide a systematic analysis of statistical contrasts and associated Bayes factors (Figures 2D and 3D). These new results allow us to directly support some of our original conclusions, but also provide a more nuanced picture. Starting from those regions where cerebral word representations are significantly predictive of comprehension (Figure 3A and B) we find that in some regions many grid points exhibit evidence for a differential contribution to auditory and visual comprehension, while in other regions (e.g. IFG, AG) many grid points exhibit evidence for no modality specificity (Figure 3D). We have revised the Discussion to fully reflect these results.

We would like to point out that our data indeed suggest the involvement of temporal regions in visual speech comprehension (right STG in Figure 3B), but that this region does not predict auditory speech comprehension. Our main conclusion is therefore that regions predictive of auditory and visual comprehension are largely distinct.

For example: “Thereby our results support a route of visual speech into auditory cortical and temporal regions but provide no evidence for an overlap of speech representations in the temporal lobe that would facilitate both lip-reading and acoustic speech comprehension.” A more elaborate summary of our conclusions can be found in the Discussion.

The same problem with interpretation of null effects arises in the cross-modality decoding analyses. It is striking that these analyses are reliable for audiovisual speech but not for auditory speech. However, while this shows that the method the authors are using can detect reliable effects of a certain size, the analyses presented cannot be used to infer whether the most likely interpretation is that cross-decoding of unimodal auditory and visual speech is absent, or that it is present, but fails to reach a stringent level of significance. The authors wish to conclude that this effect is absent, they say: "The inability to cross-classify auditory and visual speech from local brain activity further supports the conclusion that acoustic and visual speech representations are largely distinct", but do not have a sound statistical basis on which to draw this conclusion.

We addressed this comment in different ways. First, by revising the significance testing of the cross-decoding performance, for which we now report brain-wide cluster-based permutation statistics. This new statistical analysis (which is now consistent with the cluster statistics in the rest of the manuscript), did not yield any significant effects (Figure 2—figure supplement 3A). Second, we now also provide Bayes factors based on t-values, derived from a comparison with a random distribution. The topographical maps (Figure 2—figure supplement 3B) show that for the large majority of brain regions, cross-classification is not possible (Bayes factors supporting the H_0_), with the exception of some irregular grid points. An exploratory cluster analysis (only for the purpose of this response) found that, even with a minimum cluster size of 1 grid point, no significant clusters occurred. We are therefore confident that cross-classification is not possible in a meaningful way between the auditory and visual conditions for the present data.

I think some substantial additional analysis, and careful consideration of which conclusions are, or are not supported by direct statistical analyses are required if this work is to be published with the current framing. This is not to say that word identity decoding in auditory and visual speech is uninteresting, or that the seeming lack of correlation between these abilities is not of interest. Only that I found the fundamental claim in the paper – that there's no common phonological representation of auditory and visual speech – to be insufficiently supported by the data presented in the current manuscript.

The revised paper now clearly spells out which results are substantiated by appropriate statistical evidence for modality specific results, and which not. In doing so, we can now provide a more detailed and nuanced view on which regions contribute perceptually-relevant word encoding for acoustic and visual speech.

For example, in the first paragraph of the Discussion we state: “Our results show that the cerebral representations of auditory and visual speech are mediated by both modality specific and overlapping (potentially amodal) representations. While several parietal, temporal and frontal regions were engaged in the encoding of both acoustically and visually conveyed word identities (“stimulus classification”), comprehension in both sensory modalities was largely driven by distinct networks. Only the inferior frontal and angular gyrus contained regions that contributed similarly to both auditory and visual comprehension.”

Reviewer #3:In this manuscript, Keital et al. use MEG pattern classification to compare the cortical regions involved in the encoding and comprehension of auditory and visual speech. The article is well-written, addresses a scientifically valuable topic, and extends the current literature by employing multivariate approaches. In their stimulus classification analysis, the authors found that visually-presented words were best classified on the basis of signals localized to occipital regions, while auditory words were best classified in perisylvian regions. In their neuro-behavioral decoding analysis, the authors found that classification strength predicted behavioral accuracy in largely distinct regions during auditory versus visual stimulation. They conclude that perceptually-relevant speech representations are largely modality specific.While I largely agreed with the authors' rationale in employing these techniques, I had some concerns regarding the statistical/mathematical details of their approach. First and most importantly, the neuro-behavioral decoding analysis presented in Figure 3A and 3C does not directly test the hypothesis that forms their primary conclusion (i.e., that "Largely distinct networks mediate perceptually-relevant auditory and visual speech representations"). To test this hypothesis directly, it would be necessary to compare neuro-behavioral decoding accuracy between the auditory and visual conditions for each vertex and then multiple-comparison correct across vertices. That is, rather than comparing decoding accuracy against the null separately for each condition, decoder accuracy should be compared directly across conditions. The authors perform a similar analysis in Figure 3D, but this analysis may inflate Type-I error because it involves pre-selecting peaks identified in each single-condition analysis.

As reported in the reply to reviewer 2, we now provide direct statistical contrasts between A and V conditions and report Bayes factors for the null hypotheses to support our conclusions. These analyses in large support our previous claims, but also highlight a more nuanced picture, as reflected in the revised Discussion. Please see also comments above. e.g. in the first paragraph of the Discussion we now write: “Our results show that the cerebral representations of auditory and visual speech are mediated by both modality-specific and overlapping (potentially amodal) representations. While several parietal, temporal and frontal regions were engaged in the encoding of both acoustically and visually conveyed word identities (“stimulus classification”), comprehension in both sensory modalities was largely driven by distinct networks. Only the inferior frontal and angular gyrus contained regions that contributed similarly to both auditory and visual comprehension.”

Second, throughout the manuscript, a wide variety of statistical techniques are used which are sometimes internally inconsistent, weakly or not explicitly justified, or interpreted in a manner that is not fully consistent with their mathematical implications. For example, for different analyses, different multiple comparison correction procedures are used (FDR vs. cluster-based FWE control), without a clear justification. It would be best to use similar corrections across analyses to ensure similar power. Also, the authors report that they: "Tested a number of different classifiers, […] then selected a correlation-based nearest-neighbour classifier as this performed slightly better than the others". Absent an a priori justification for the chosen approach, it would be helpful to know whether the reported results are robust to these design decisions.

We have revised all statistical analyses and now use the same correction methods for all fullbrain analyses. In a preliminary analysis we had tested different classifiers in their ability to classify the stimulus set (e.g. a nearest neighbour classifier based on the Euclidean distance, and correlation decoders based on different temporal sampling). Based on the overall performance across all three conditions (A,V,AV) we decided on the classifier and parameter used, although we note that the differences between version of the classifier were small (23%). In part, we now directly address this point by reporting how the stimulus classification performance depends on the spatial searchlight and the temporal resolution. We have repeated the stimulus classification using a range of spatial and temporal parameters for the searchlight. In particular, we removed the spatial information and systematically reduced the temporal resolution.

These complementary results are now mentioned in the manuscript.

“We also probed classification performance based on a number of spatio-temporal searchlights, including temporal binning of the data at 3.3, 20, 40 and 60ms, and including each neighbouring spatial grid point into the searchlight or averaging across grid points. Comparing classification performance revealed that in particular the auditory condition was sensitive to the choice of searchlight. Classification performance dropped when ignoring the spatial configuration or sampling the data at a resolution lower than 20 ms (median performance of the 10% grid points with highest performance based on each searchlight: 27.0%, 26.9% 26.5% and 25.8% for 3.3, 20, 40 and 60-ms bins and the full spatial searchlight; and 26.4% at 20ms and ignoring the spatial pattern). We hence opted for a classifier based on the source data represented as spatial searchlight (1.2-cm radius) and sampled at 20-ms resolution.”

In Author response image 1 is an illustration of these results.

**Author response image 1. sa2fig1:** 

Finally, it is not clear to me that the neuro-behavioral decoding technique employed here is particularly well-suited for identifying regions that represent participants' percepts. It seems the most natural way to perform such an analysis would be to train a classifier to predict participants' trial-wise responses. By contrast, the technique employed here compares (binary) trial-wise accuracy with "representational distance", computed as the difference between "the average correlation with trials of the same (correct) word identity and the mean of the correlation with the three alternatives." Thus, the classifier would not be expected to identify regions with response patterns that predict participants' percepts, but those which exhibit a target-like pattern when the participant responds accurately. It is therefore perhaps best conceived of as an "accuracy classifier" rather than a "percept classifier." This may be problematic for the authors' interpretation because activity in areas unrelated to perceptual representations (e.g., areas involved in unimodal attention) could also be predictive of accuracy.

The reviewer touches on an important issue concerning the interpretation of the mapped representations. Our study was motivated by the notion of intersection information, that is the search for cerebral representations of a stimulus that are used for the respective single trial behaviour (Panzeri et al., 2017; Pica et al., 2017). This intersection information can be formalised theoretically and can be measured using information theoretic approaches. However, these principled approaches are still computationally inefficient. Following the neuroimaging field, we hence opted for a distanceto-bound method, where the amount of sensory evidence captured in a classifier is regressed against behaviour using a linear model (see, e.g. Grootswagers, Cichy and Carlson, 2018). This method is computationally cheaper and allows for full-brain permutation statistics. In contrast to this approach, a classifier trained on participants response (choice) would, at least on error trials, possibly capture representations not related to speech encoding but relevant for behaviour only (e.g. pre-motor activity). These could be e.g. brain activity that leads to perceptual errors based on intrinsic fluctuations in neural activity in sensory pathways, noise in the decision process favouring one alternative response among four choices, or even noise in the motor system that leads to a wrong button press without having any relation to sensory representations at all. The latter example highlights the need to base any analysis of behaviourally-relevant sensory representations (i.e. the intersection information) on classifiers which are firstly trained to discriminate the relevant sensory information and which are then probed as to how behaviourally-relevant the classifier output is. We have revised the Discussion to better explain the rationale of our approach in this respect.

Finally, in some cases, the methods description was not self-sufficient, leaving the reader to consult references to fully understand. One critical question is how the spatial and temporal dimensions of the data were used in the classifier. If classification is primarily driven by the temporal dimension, it is not clear that successful classification really relies on pattern similarity in population responses, rather than inter-trial temporal covariation produced by, e.g., phase-resetting or entrainment to stimulus dynamics. In the cited methods articles, spatial classification is performed separately for different time bins, alleviating this concern. It would be important to critically consider this detail in interpreting these results.

We have revised the Materials and methods in many instances for ensure that all procedures are described clearly. We have added more information about the spatio-temporal dimensions of the classifier (see also above), noting that both spatial and temporal patterns of local brain activity were contributing to the stimulus classification. Whether the neural “mechanisms” mentioned, such as phase-resetting, indeed contribute to the cerebral representations studied here is a question that is surely beyond the scope of this study.

[Editors’ note: what follows is the authors’ response to the second round of review.]

Essential revisions:1) Brain-wide, multiple-comparison corrected tests comparing auditory versus visual decoding are still lacking. The authors have now provided vertex-wise Bayes factors within areas that showed significant decoding in each individual condition. Unfortunately, this is not satisfactory, because these statistics are (1) potentially circular because ROIs were pre-selected based on an analysis of individual conditions, (2) not multiple-comparison corrected, and (3) rely on an arbitrary prior that is not calibrated to the expected effect size. Still, ignoring these issues, the only area that appears to contain vertices with "strong evidence" for a difference in neuro-behavioral decoding is the MOG, which wouldn't really support the claim of "largely distinct networks" supporting audio vs. visual speech representation.The authors may address these issues, for instance, byi) presenting additional whole-brain results – e.g. for a direct comparison of auditory and visual classification (in Figure 2) and of perceptual prediction (in Figure 3).ii) presenting voxel-wise maps of Bayesian evidence values (as in Figure 2—figure supplement 3) for the statistical comparisons shown in Figure 2D, and Figure 3Diii) in the text included in Figure 2D and 3D making clear what hypotheses correspond to the null hypothesis and to the alternative hypothesis (i.e. auditory = visual, auditory <> visual).

We addressed this comment using additional data analysis to ensure that all claims are supported by sufficient statistical evidence.

i) In the revised manuscript, we now provide the suggested full-brain cluster-corrected contrasts between auditory and visual conditions for both main analyses (Figure 2—figure supplement 1A, Figure 3—figure supplement 1A). However, we caution against the interpretation of condition-wise differences at grid points that do not exhibit significant evidence for the primary “function” of interest. We therefore refrain from interpreting condition-wise differences in word classification performance (or the prediction of comprehension) at grid points that do not exhibit significant word classification (prediction of comprehension) in at least one condition (vs. the randomization null). Hence, we masked the full-brain cluster-corrected condition differences against all grid points contributing to word classification (or the prediction of comprehension) in at least one modality (while also reporting all clusters in the figure caption).

Importantly, and in line with the general concerns raised about interpreting null results, these full-brain condition differences can provide evidence in favour of a condition difference, and therefore support the existence of modality specific regions. However, they cannot provide evidence in favour of a null finding of no modality specialisation. The analysis of Bayes factors provided in the previous revision, which we retained in Figures 2D,3D, in contrast, can provide such evidence in favour of a null result. Hence in the revised manuscript we kept the Bayes factors in the main figure, while now also providing the full-brain cluster-based statistics in the supplemental material. Importantly, we base all interpretations of the results on the combined evidence provided by the full-brain condition differences and these Bayes factors.

ii) We now also provide full-brain maps with Bayes factors for contrasts between the two modalities (in Figure 2—figure supplement 1B, Figure 3—figure supplement 1B, alongside the full-brain cluster maps).

iii) We have added the specific hypotheses to Figures 2 and 3 as suggested.

The results of these additional statistical tests do not affect our main findings, but they support the conclusions derived from the ROI-based analysis. We carefully ensured that the revised manuscript clearly acknowledges that the individual analyses are inconclusive for some parts of the brain or offer specific evidence for no difference between modalities in other parts. For example:

“A separate full-brain cluster-based permutation test (Figure 3—figure supplement 1A) provided evidence for a significant modality specialisation for auditory words in four clusters in the left middle occipital gyrus, left calcarine gyrus, right posterior angular gyrus, and bilateral supplementary motor area. The corresponding full-brain Bayes factors (Figure 3—figure supplement 1B) support this picture but also provide no evidence for a modality preference, or inconclusive results, in many other regions.”

To acknowledge that there is a large number of grid points that do not show a modality difference, we have also revised the title of this manuscript to “Shared and modality-specific brain regions that mediate auditory and visual word comprehension”

The comment also suggests that the analyses of ROI-specific comparisons may be circular. First, we now provide the full brain results for the Bayes factors in Figure 2—figure supplement 1B and Figure 3—figure supplement 1B. Concerning the ROI-based results in Figures 2,3, it is important to note that we compare condition-wise differences (as Bayes factors) within regions pre-selected to show an effect in at least one of the two modalities, hence independent of the contrast. We do so, as the interpretation of a condition-wise difference (e.g. in word classification) in a brain region not exhibiting significant classification in any condition is difficult, if not impossible. Such pre-selection of electrodes or regions of interest is very common (Cheung et al., 2016, Giordano et al., 2017, Karas et al., 2019, Mihai et al., 2019, Ozker, Yoshor and Beauchamp, 2018) and supported by the use of orthogonal contrasts for selection and comparison.

The comment further notes that we relied on an “arbitrary prior” to calculate Bayes factors. We would like to emphasize that the JZS Bayes factor, while not being data-driven by the present study, is not arbitrary. It has been advocated heavily by a number of studies for situations where no specific information about the expected effects sizes is available 2009(Rouder et al., 2012; ) and is highly accepted in the current literature (Guitard and Cowan, 2020, Mazor, Friston and Fleming, 2020, Puvvada and Simon, 2017, Kimel, Ahissar and Lieder, 2020). The default scale of 22 corresponds to the assumption that the prior of effect sizes follows a Cauchy distribution with 50% of probability mass placed on effect sizes smaller than 22, and 50% larger than this number (Schönbrodt and Wagenmakers, 2018). Looking at our data, the effect sizes in the neurobehavioral analysis (Table 2) seem to follow such a pattern rather well (4 out of 9 effect sizes, calculated as Cohen’s D, are smaller than 22, while 5 are larger than this). Hence, taking this one specific statistical contrast from our study as evidence, it makes sense to assume that similar effect sizes are expected also in the other tests, such as the condition-wise differences for which we report the Bayes factors in Figures 2D, 3D. Of course, in an ideal case one would base the choice of the prior on pre-existing and independent data, however, unfortunately such data were not available.

2) As noted before by reviewer 3, the classifiers used in this study do not discriminate between temporal versus spatial dimensions of decoding accuracy. This leaves it unclear whether the reported results are driven by (dis)similarity of spatial patterns of activity (as in fMRI-based MVPA), temporal patterns of activity (e.g., oscillatory "tracking" of the speech signal), or some combination. As these three possibilities could lead to very different interpretations of the data, it seems critical to distinguish between them. For example, the authors write "the encoding of the acoustic speech envelope is seen widespread in the brain, but correct word comprehension correlates only with focal activity in temporal and motor regions," but, as it stands, their results could be partly driven by this non-specific entrainment to the acoustic envelope.In their response, the authors show that classifier accuracy breaks down when spatial or temporal information is degraded, but it would be more informative to show how these two factors interact. For example, the methods article cited by the authors (Grootswagers, Wardle and Carlson, 2017) shows classification accuracy for successive time bins after stimulus onset (i.e., they train different classifiers for each time bin 0-100 ms, 100-200 ms, etc.). The timing of decoding accuracy in different areas could also help to distinguish between different plausible explanations of the results.

This comment raises a number of interesting questions, which we (partly) addressed in the previous round. Unfortunately, maybe, our treatment of this question in the previous round was not fully comprehensive, and the results were reported only in a single sentence in the Materials and methods. To address these points in full, we have done a series of additional analyses and report these results more extensively in the manuscript and in this reply.

We start noting that the use of a spatio-temporal searchlight in MEG source analysis is common in the literature (Cao et al., 2019, Giordano, et al., 2017, Kocagoncu et al., 2017, Su et al., 2012), and is analogous to the inclusion of all M/EEG sensors in sensory-based analyses relying on classification methods or RSA analysis (Cichy and Pantazis, 2017, Guggenmos, Sterzer and Cichy, 2018, Kaiser, Azzalini and Peelen, 2016 ). However, and maybe unlike the use of spatial searchlights in many fMRI studies, the spatial component (on the scale of 1.2 cm as used here) in MEG source space is expected to add only minor information, given the natural smoothness of MEG source data. Hence, the emphasis for the present analysis was on the temporal domain.

Concerning the duration of the chosen time window in the main analysis (500ms), we note that this has been adapted to the specifics of the experimental design and task (Figure 1). In particular, this window was chosen to cover the different target words as much as possible without including the following word in the sentence (which is different in every sentence and would therefore have contaminated the decoding analysis). Spoken words are temporally extended, and to study their cerebral encoding it makes sense to use a time window that covers most of the stimulus duration. Otherwise one runs the risk of capturing processes related to lexical exploration/competition or word predictions (Klimovich-Gray et al., 2019, Kocagoncu, et al., 2017, Marslen-Wilson and Welsh, 1978). We therefore chose a longer time window. This is possibly in contrast to studies on visual object encoding, where stimuli are often flashed for a few tens of milliseconds only, and the encoding window often extends this by a certain, but ambiguous amount. In contrast, the choice of a 500-ms window here is parsimonious given the nature of the stimuli and task.

Importantly, by task design, the sentence continued beyond the target word, which was the 2nd or 3rd last word in the sentence (Figure 1). Hence, the sentence stimuli continued beyond the analysis window (for 1.48 s to 2.81s (1.98 ± 0.31 s [*M* ± *SD*])) and any motor response, or the relevant response options presented to the participants, followed much later. Using this design, we ensured that motor preparation is very unlikely to emerge within the analysed time window.

To address this comment using data analysis, we first compared the classification performance with and without the spatial dimension in the searchlight. As noted above, the natural expectation is that the spatial dimension adds only little additional information in comparison to the time domain. In the additional analysis, we quantified whether and by how much the spatial dimension adds to the ability to classify word identities (Author response image 2). The average classification performance (within the 10% grid points with the highest performance in the classification including searchlight) differed little (auditory: 27.19 ± 0.48% [M ± SD] with searchlight vs 26.72 ± 1.89% without searchlight; visual: 28.71 ± 1.55% with searchlight vs 27.71 ± 2.44% without searchlight, [averages across grid points and participants]; this equals an overall percent change in performance of -1.49% and -2.45% in the auditory and visual condition, respectively). A correlation between the full-brain maps with and without spatial dimension showed that 99.8% of grid points in the auditory, and 99.7% of grid points in the visual condition, were significantly correlated (Pearson correlation, at p <.05, FDR-corrected). Finally, a direct group-level comparison revealed that the majority of grid points (66.9% in the auditory and 63.9% in the visual condition) showed evidence for no difference (i.e. BF_10_ < 1/3) between including or excluding the spatial dimension (see Author response image 2, bottom panel). This shows that the spatial component contributes only modestly to the classification performance. This result is now reported in the manuscript as follows:

“We also probed the influence of the spatial searchlight by i) including each neighbouring spatial grid point into the searchlight, or ii) averaging across grid points, and iii) by not including a searchlight altogether. Ignoring the spatial pattern by averaging grid points led to a small drop in classification performance (individual vs average grid points: auditory, 27.19 ± 0.48 vs 26.72 ± 0.67; visual, 28.71 ± 1.55 vs 27.71 ± 1.25 [M ± SD]). Performance also dropped slightly when no searchlight was included (auditory, 26.77 ± 1.93; visual, 27.86 ± 2.50 [M ± SD]).”

**Author response image 2. sa2fig2:** Word classification performance with and without a spatial searchlight. Top panel: original results including a 1.2-cm searchlight (as in Figure 2 in the manuscript). Middle panel: classification results without searchlight. Bottom panel: Bayes factors of a group-level t-test comparing classification performance with and without searchlight. The majority of grid points (66.9% in the auditory and 63.9% in the visual condition) showed evidence for no difference (i.e. BF10 < 1/3) between tests, while only a small fraction of grid points show evidence for a strong or substantial difference between test. There is no systematic improvement when including the searchlight.

To address the question of how the temporal discretisation of MEG activity within this time window affects classification performance, we compared classifiers operating on data binned at 20, 40, and 60 ms bins. These results had already been reported in the previous reply letter and were included in the Materials and methods section. To give more emphasis to these results we have moved them to a separate section (*“*Selection of parameters and classifier procedures*”*).

In brief, we find that a temporal resolution of 20 ms is sufficient to recover most of the stimulus information contained in the data. Using larger windows would lead to a loss of information, while shorter windows did not seem to add any classification performance.

To address the concern that some form of non-specific temporal entrainment of brain activity may confound our results, we implemented a further analysis using shorter time windows. We divided the original 500-ms window into shorter time epochs, as suggested by the reviewer. The length of these (140 ms) was chosen to avoid contributions of rhythmic activity at the critical time scales of 2 – 4 Hz, which have prominently been implied in speech-to-brain entrainment (e.g. Ding and Simon, 2014, Luo, Liu and Poeppel, 2010, Molinaro and Lizarazu, 2017). We repeated the word classification (and the prediction of comprehension) in 7 partly overlapping (by 60 ms) epochs of 140 ms duration. We then subjected the epoch-specific results to the same full-brain cluster-based permutation statistics as used for the full 500-ms window. Finding significant word classification (or prediction of comprehension) in these shorter epochs would speak against the notion that some sort of entrainment critically contributes to (or confounds) our results.

Before presenting this result, we note that introducing 7 additional time epochs adds to the problem of correcting for multiple comparisons. The results of such a fine-grained analysis can either be considered at a much more stringent criterion (when correcting across all 7 time epochs) or a less stringent criterion (when not correcting, and hence accepting a sevenfold higher false positive rate) compared to the main analysis in the manuscript. We here chose to correct for multiple tests by using Bonferroni correction and adjusting the significance threshold by ɑ=0.057=0.0071.

Author response image 3 presents a cumulative whole-brain histogram of the significant grid points across all 7 epochs and the epoch-specific number of grid points that are significant in each epoch.

**Author response image 3. sa2fig3:** Classification performance and neurobehavioural prediction over time. Top panels represent cumulative whole-brain histograms of the significant grid points across all 7 epochs and bottom panels represent the epoch-specific number of grid points that are significant in each epoch. Please note that these results are very conservative due to the Bonferroni-corrected threshold of ɑ = 0.0071. The maps resemble those of the full-window analysis presented in the manuscript.

These results provide several important insights: First, they confirm that significant word classification (and prediction of comprehension) can be obtained in shorter epochs, arguing against temporally entrained brain activity presenting some critical confounding factor. Second, the results show that the significant grid points obtained (cumulatively) across epochs cover largely the same regions found using the full time window. Out of those grid points reported in Figures 2 and 3, the percentage of grid points that becomes significant in at least one time epoch is: *forword classification*: auditory 44.7% of grid points, visual 60.3%; for the *neurobehavioural analysis*: auditory 19.2% of grid points, visual 24.4%. Note that without the very strict Bonferroni-correction, a much larger frequency of original grid points is also found in the short time windows (*classification*: 77.3%/82.3% and *neurobehavioural*: 57.9%/77.2%; auditory/visual).

Third, the fraction of grid points being significant in at least one time epoch but not significant in the analysis of the full time window, and hence emerging *only* in the analysis of shorter time epochs, is small: 9.2%/5.2% for auditory/visual *word classification*, 4.6%/2.7% for auditory/visual *neurobehaviouralprediction*.

These results suggest that the use of the full 500-ms time window can be justified: first, this time window covers a large proportion of target words relevant for the behavioural task and is therefore directly motivated given the experimental design. Second, it is sufficiently separated from the motor response (c.f. Figure 1 and Materials and methods). Third, it is not confounded by temporally entrained activity at 2 – 4 Hz (Author response image 3). And finally, the use of shorter time epochs reveals largely the same brain regions (Author response image 3). However, in contrast to the full time window, the length of any shorter window will always remain arbitrary, hence necessitating one more arbitrary choice in the analysis pipeline.

Finally, it is somewhat unclear how spatial and temporal information are combined in the current classifier. Figure 1—figure supplement 2 creates the impression that the time-series for each vertex within a spotlight were simply concatenated. However, this would conflate within-vertex (temporal) and across-vertex (spatial) variance.

As is common in classification or RSA analyses based on spatio-temporal activity patterns, we indeed concatenated the time series obtained at the different grid points within each spatial neighbourhood. The same procedure is often used in studies using fMRI voxel based analysis and MEG/EEG source level analyses (Cao, et al., 2019, Giordano, et al., 2017, Kocagoncu, et al., 2017, Su, et al., 2012). While this indeed conflates spatial and temporal information, we note that the 1.2-cm radius of the spatial searchlight still retains a high level of spatial information in the overall source map. Most importantly, and as shown by the above analyses, only a small amount of extra information is contained in the spatial pattern, and hence this mixing of spatial and temporal dimensions did not influence our results heavily.

3) The concern that the classifier could conceivably index factors influencing "accuracy" rather than the perceived stimulus does not appear to be addressed sufficiently. Indeed, the classifier is referred to as identifying "sensory representations" throughout the manuscript, when it could just as well identify areas involved in any other functions (e.g., attention, motor function) that would contribute to accurate behavioral performance. This limitation should be acknowledged in the manuscript. The authors could consider using the timing of decoding accuracy in different areas to disambiguate these explanations.

We agree that other factors enhancing the cerebral processes involved in encoding the sensory input and translating this into a percept (e.g. attention) could affect behaviour. At the same time, we designed the paradigm to specifically avoid confounding factors, such as motor preparation, by placing the target words not at the end of the sentence (c.f. Figure 1; Materials and methods). In the revised manuscript we discuss this as follows:

“While we found strongest word classification performance in sensory areas, significant classification also extended into central, frontal and parietal regions. This suggests that the stimulus-domain classifier used here may also capture processes potentially related to attention or motor preparation. While we cannot rule out that levels of attention differed between conditions, we ensured by experimental design that comprehension performance did not differ between modalities. In addition, the relevant target words were placed not at the end of the sentence to prevent motor planning and preparation during their presentation (see Stimuli).”

The idea to investigate the time courses within specific brain areas is interesting, but opens a very large number of additional degrees of freedom. The above presented analysis of small time-windows did this to some extent. We would like to refrain from a complete analysis of decoding over time in different brain areas, as this is a different research question than the one the study was designed to answer.

The authors state in their response that classifying based on the participant's reported stimulus (rather than response accuracy) could "possibly capture representations not related to speech encoding but relevant for behaviour only (e.g. pre-motor activity). These could be e.g. brain activity that leads to perceptual errors based on intrinsic fluctuations in neural activity in sensory pathways, noise in the decision process favouring one alternative response among four choices, or even noise in the motor system that leads to a wrong button press without having any relation to sensory representations at all."But, it seems that all of these issues would also effect the accuracy-based classifier as well. Moreover, it seems that intrinsic fluctuations in sensory pathways, or possibly noise in the decision process, are part of what the authors are after. If noise in a sensory pathway can be used to predict particular inaccurate responses, isn't that strong evidence that it encodes behaviorally-relevant sensory representations? For example, intrinsic noise in V1 has been found to predict responses in a simple visual task in non-human primates, with false alarm trials exhibiting noise patterns that are similar to target responses (Seidemann and Geisler 2018). Showing accurate trial-by-trial decoding of participants' incorrect responses could similarly provide stronger evidence that a certain area contributes to behavior.

Unfortunately, we are not sure what “accuracy-based classifier” here refers to. The classifier used in the present study classifies the word “identity”, and operates in the stimulus domain, not in the domain of participants' response or accuracy. The evidence contained in this stimulus-domain classifier is then used, in a regression model, as a predictor of participants' response accuracy. In the response to the previous comments, we contrasted this approach with a putative analysis based on a classifier directly trained on participant’s choice; such an analysis had been mentioned in the previous set of reviewer comments (“It seems the most natural way to perform such an analysis would be to train a classifier to predict participants' trial-wise responses” to quote from the previous set of comments). We have tried to make our approach clearer early in the manuscript:

“Using multivariate classification, we quantified how well the single-trial identity of the target words (18 target words, each repeated 10 times) could be correctly predicted from source-localised brain activity (“stimulus classifier”).”

As we argued previously, we believe that our analysis has several strengths in contrast to this suggestion of a choice-based classifier. In particular, the argument that “these issues” (quoted from this reviewer’s point above) affect our analysis as well, does not seem clear to us. For example, a classifier trained on choice would contain information about activity patterns that drive behaviour (erroneously) on trials where participants missed the stimulus. In this case, their sensory cortices would not encode these and overt behaviour would be driven by noise somewhere in the decision or motor circuits. In contrast, an analysis capitalising first on the encoding of the relevant sensory information (by a stimulus-domain classifier), and then using a signature of how well this is encoded in cerebral activity, avoids being confounded by decision or motor noise. The reasoning behind our approach is very much in line with recent suggestions made by various groups for how to best elucidate the sensory representations that drive perception and comprehension, based on an analysis that in a first stage chiefly capitalises on the encoding of sensory information (Grootswagers, Cichy, and Carlson, 2018, Panzeri et al., 2017) and then relates this to behavioural performance. We refined the manuscript to ensure this is very clearly spelled out in different places, for example:

“This approach directly follows the idea to capture processes related to the encoding of external (stimulus-driven) information and to then ask whether these representations correlate over trials with the behavioural outcome or report.”

Of course, no technical approach is perfect, and the statement about noise in a sensory pathway touches upon an interesting issue (“If noise in a sensory pathway can be used to predict particular inaccurate responses, isn't that strong evidence that it encodes behaviourally-relevant sensory representations?” quoted from the present set of comments). Indeed, for our approach it does not matter where precisely variations in noise in the encoding process emerge, as long these affect either the cerebral reflection of sensory information or how this relates on a trial-by-trial basis to behaviour. However, by design of our paradigm (4 alternative response choices) and the type of classifier used (c.f. Materials and methods), the present analysis is capitalising on the correct encoding of the sensory information, while we cannot specifically link the incorrect encoding of a stimulus with behaviour. This arises because evidence against the correct stimulus is not directly evidence in favour of one specific other stimulus; this is in contrast to studies using a two-alternative forced choice design, where evidence against one stimulus / response option directly translates into evidence in favour of the other. We have revised the text to directly reflect this limitation of the present approach:

“In addition, by design of our experiment (4 response options) and data analysis, the neurobehavioral analysis was primary driven by trials in which the respective brain activity encoded the sensory stimulus correctly. We cannot specifically link the incorrect encoding of a stimulus with behaviour. This is in contrast to studies using only two stimulus or response options, where evidence for one option directly provides evidence against the other (Frühholz et al., 2016, Petro et al., 2013).”

**References**

Cao, Y., Summerfield, C., Park, H., Giordano, B. L., and Kayser, C. (2019). Causal inference in the multisensory brain. Neuron, 102(5), 1076-1087. e1078.

Cheung, C., Hamilton, L. S., Johnson, K., and Chang, E. F. (2016). The auditory representation of speech sounds in human motor cortex. eLife, 5, e12577.

Cichy, R. M., and Pantazis, D. (2017). Multivariate pattern analysis of MEG and EEG: A comparison of representational structure in time and space. NeuroImage, 158, 441-454.

Ding, N., and Simon, J. Z. (2014). Cortical entrainment to continuous speech: functional roles and interpretations. Front Hum Neurosci, 8, 311. doi: 10.3389/fnhum.2014.00311

Frühholz, S., Van Der Zwaag, W., Saenz, M., Belin, P., Schobert, A.-K., Vuilleumier, P., and Grandjean, D. (2016). Neural decoding of discriminative auditory object features depends on their socio-affective valence. Social cognitive and affective neuroscience, 11(10), 1638-1649.

Giordano, B. L., Ince, R. A. A., Gross, J., Schyns, P. G., Panzeri, S., and Kayser, C. (2017). Contributions of local speech encoding and functional connectivity to audio-visual speech perception. elife, 6. doi: 10.7554/eLife.24763

Grootswagers, T., Cichy, R. M., and Carlson, T. A. (2018). Finding decodable information that can be read out in behaviour. Neuroimage, 179, 252-262.

Guggenmos, M., Sterzer, P., and Cichy, R. M. (2018). Multivariate pattern analysis for MEG: a comparison of dissimilarity measures. Neuroimage, 173, 434-447.

Guitard, D., and Cowan, N. (2020). Do we use visual codes when information is not presented visually? Memory and Cognition.

Haxby, J. V., Connolly, A. C., and Guntupalli, J. S. (2014). Decoding neural representational spaces using multivariate pattern analysis. Annual review of neuroscience, 37, 435-456.

Kaiser, D., Azzalini, D. C., and Peelen, M. V. (2016). Shape-independent object category responses revealed by MEG and fMRI decoding. Journal of neurophysiology, 115(4), 2246-2250.

Kamitani, Y., and Tong, F. (2005). Decoding the visual and subjective contents of the human brain. Nature neuroscience, 8(5), 679-685.

Karas, P. J., Magnotti, J. F., Metzger, B. A., Zhu, L. L., Smith, K. B., Yoshor, D., and Beauchamp, M. S. (2019). The visual speech head start improves perception and reduces superior temporal cortex responses to auditory speech. eLife, 8.

Kimel, E., Ahissar, M., and Lieder, I. (2020). Capacity of short-term memory in dyslexia is reduced due to less efficient utilization of items' long-term frequency. bioRxiv.

Klimovich-Gray, A., Tyler, L. K., Randall, B., Kocagoncu, E., Devereux, B., and Marslen-Wilson, W. D. (2019). Balancing prediction and sensory input in speech comprehension: The spatiotemporal dynamics of word recognition in context. Journal of Neuroscience, 39(3), 519-527.

Kocagoncu, E., Clarke, A., Devereux, B. J., and Tyler, L. K. (2017). Decoding the cortical dynamics of sound-meaning mapping. Journal of Neuroscience, 37(5), 1312-1319.

Luo, H., Liu, Z., and Poeppel, D. (2010). Auditory cortex tracks both auditory and visual stimulus dynamics using low-frequency neuronal phase modulation. PLoS Biol, 8(8), e1000445. doi: 10.1371/journal.pbio.1000445

Marslen-Wilson, W. D., and Welsh, A. (1978). Processing interactions and lexical access during word recognition in continuous speech. Cognitive psychology, 10(1), 29-63.

Mazor, M., Friston, K. J., and Fleming, S. M. (2020). Distinct neural contributions to metacognition for detecting, but not discriminating visual stimuli. eLife, 9, e53900.

Mihai, P. G., Moerel, M., de Martino, F., Trampel, R., Kiebel, S., and von Kriegstein, K. (2019). Modulation of tonotopic ventral medial geniculate body is behaviorally relevant for speech recognition. eLife, 8.

Molinaro, N., and Lizarazu, M. (2017). Delta (but not theta)-band cortical entrainment involves speech-specific processing. European Journal of Neuroscience, 48(7), 2642-2650. doi: doi:10.1111/ejn.13811

Ozker, M., Yoshor, D., and Beauchamp, M. S. (2018). Frontal cortex selects representations of the talker’s mouth to aid in speech perception. eLife, 7, e30387.

Panzeri, S., Harvey, C. D., Piasini, E., Latham, P. E., and Fellin, T. (2017). Cracking the neural code for sensory perception by combining statistics, intervention, and behavior. Neuron, 93(3), 491-507.

Petro, L. S., Smith, F. W., Schyns, P. G., and Muckli, L. (2013). Decoding face categories in diagnostic subregions of primary visual cortex. European Journal of Neuroscience, 37(7), 1130-1139.

Puvvada, K. C., and Simon, J. Z. (2017). Cortical representations of speech in a multitalker auditory scene. Journal of Neuroscience, 37(38), 9189-9196.

Ritchie, J. B., Kaplan, D. M., and Klein, C. (2019). Decoding the brain: Neural representation and the limits of multivariate pattern analysis in cognitive neuroscience. The British Journal for the Philosophy of Science, 70(2), 581-607.

Ritchie, J. B., Tovar, D. A., and Carlson, T. A. (2015). Emerging object representations in the visual system predict reaction times for categorization. PLoS computational biology, 11(6), e1004316.

Rouder, J. N., Morey, R. D., Speckman, P. L., and Province, J. M. (2012). Default Bayes factors for ANOVA designs. Journal of Mathematical Psychology, 56(5), 356-374.

Rouder, J. N., Speckman, P. L., Sun, D., Morey, R. D., and Iverson, G. (2009). Bayesian t tests for accepting and rejecting the null hypothesis. Psychonomic bulletin and review, 16(2), 225-237.

Schönbrodt, F. D., and Wagenmakers, E.-J. (2018). Bayes factor design analysis: Planning for compelling evidence. Psychonomic bulletin and review, 25(1), 128-142.

Su, L., Fonteneau, E., Marslen-Wilson, W., and Kriegeskorte, N. (2012). Spatiotemporal searchlight representational similarity analysis in EMEG source space. Paper presented at the 2012 Second International Workshop on Pattern Recognition in NeuroImaging.